

# Towards monitoring $CO_2$ source-sink distribution over India via inverse modelling: Quantifying the fine-scale spatiotemporal variability of atmospheric $CO_2$ mole fraction

Vishnu Thilakan[1,4], Dhanyalekshmi Pillai[1,4], Christoph Gerbig[2], Michal Galkowski[2,3], Aparnna Ravi[1,4], and Thara Anna Mathew[1]

[1]Indian Institute of Science Education and Research Bhopal (IISERB), Bhopal, India

[2]Max Planck Institute for Biogeochemistry, Jena, Germany

[3]AGH University of Science and Technology, Kraków, Poland

[4]Max Planck Partner Group (IISERB), Max Planck Society, Munich, Germany

*Correspondence to*: Dhanyalekshmi Pillai (dhanya@iiserb.ac.in, kdhanya@bgc-jena.mpg.de)

**Abstract**

Improving the estimates of $CO_2$ sources and sinks over India through inverse methods calls for a comprehensive
atmospheric monitoring system involving atmospheric transport models that realistically account for atmospheric $CO_2$ variability along with good coverage of ground-based monitoring stations. This study investigates the importance of representing fine-scale variability of atmospheric $CO_2$ in models for the optimal use of observations through inverse modelling. The unresolved variability of atmospheric $CO_2$ in coarse models is quantified by using WRF-Chem simulations at a spatial resolution of 10 km × 10 km. We show that the
representation errors due to unresolved variability in the coarse model with a horizontal resolution of one degree (~ 100 km) are considerable (median values of 1.5 ppm and 0.4 ppm for the surface and column $CO_2$, respectively) compared to the measurement errors. The monthly averaged surface representation error reaches up to ~5 ppm, which is comparable to a quarter to half of the magnitude of seasonal variability. Representation error shows a strong dependence on multiple factors such as time of the day, season, terrain heterogeneity, and
changes in meteorology and surface fluxes. By employing a first-order inverse modelling scheme using pseudo observations from nine tall tower sites over India, we show that the Net Ecosystem Exchange (NEE) flux uncertainty solely due to unresolved variability is in the range of 3.1 to 10.3% of the total NEE of the region. By estimating the representation error and its impact on flux estimations during different seasons, we emphasize the need for taking account of fine-scale $CO_2$ variability in models over the Indian subcontinent to better understand
processes regulating $CO_2$ sources and sinks. The efficacy of a simple parameterization scheme is further demonstrated to capture these unresolved variations in coarse models.

## 1 Introduction

Accurate assessment of sources and sinks of $CO_2$ is essential in planning and implementing mitigation strategies for greenhouse gas emissions and associated climate change. However, estimations of $CO_2$ fluxes contain
significant uncertainties, which increase even more with finer spatial scales such as those required for the climate change mitigation policies at regional and national levels (e.g., Ciais et al., 2014; Li et al., 2016; Cervarich et al., 2016). By using atmospheric $CO_2$ concentration measurements, the $CO_2$ fluxes can be estimated by a multi-constrained observation-modelling approach, often referred to as top-down approach or inverse



modelling (Enting, 2002). For about two decades, these top-down approaches have been widely used to understand the modifications in the carbon cycle through natural and anthropogenic induced environmental changes (Bousquet, 2000; Schimel et al., 2001; Rödenbeck et al., 2003; Patra et al., 2005). In addition to the observations, the inverse modelling system makes use of an atmospheric transport model (forward model), which determines the distribution of $CO_2$ concentration. Thereby, the inverse optimization approach derives the surface fluxes that are consistent with measured concentration. The United Nations Framework Convention on Climate Change (UNFCCC) has acknowledged the increasing capability of inverse modelling to systematically monitor greenhouse gas (GHG) concentrations (Bergamaschi et al., 2018).

Most of the inverse modelling systems rely on global atmospheric transport models with coarse horizontal resolution (often greater than one degree) (Rödenbeck et al., 2003; Peters et al., 2007; Rödenbeck et al., 2018a, b; Inness et al., 2019). These global data assimilation systems play an important role in studying continental or sub-continental fluxes at annual or sub-annual scales. However, regional estimation of fluxes using global models is hindered by the inability of these transport models to represent the observed $CO_2$ variability. The observed variability, as seen from the spatial and temporal distribution of atmospheric $CO_2$, is highly correlated with the space and time scales of weather systems (Parazoo et al., 2011). This explains the presence of large model-data mismatches in regions where mesoscale circulation is predominant (Ahmadov et al., 2007). Wind speed, wind direction, and height of the planetary boundary layer (PBL) are the critical variables that determine the atmospheric $CO_2$ variability. Strong wind normalizes other small-scale variations in observed concentration due to mixing, and the predictability can be higher during these conditions (Sarrat et al., 2007). The height of the PBL is an essential variable since the atmospheric $CO_2$ is subjected to rapid mixing up to this altitude. Hence, for a given location with a negative gradient in $CO_2$ vertical distribution, an overestimation of PBL height leads to an underestimation of $CO_2$ concentration and vice-versa (Gerbig et al., 2008).

Another important variable that impacts the $CO_2$ variability is the heterogeneous topography. Variations in topography influence the transport of the tracers. When the small-scale orographic details are not adequately represented in the models, they can lead to representation errors in $CO_2$ simulations as large as 3 ppm at scales of 100 km (Tolk et al., 2008; Pillai et al., 2010). Horizontal gradients in $CO_2$ concentrations can go up to values of 30 ppm within a spatial scale of 200 km, depending on the land surface heterogeneity (van der Molen and Dolman, 2007). Further, variations in land use patterns between neighbouring regions can cause considerable variability in the $CO_2$ surface fluxes. Thus, a proper representation of land use patterns is also important in terms of simulating $CO_2$ variability. Previous studies based on airborne measurements reported that transport models need a spatial resolution smaller than 30 km to be able to represent $CO_2$ spatial variability in the continental boundary layer (Gerbig et al., 2003). Significant efforts have been invested in deriving fluxes by taking into account these fine-scale variations (e.g., Gerbig et al., 2003; Lauvaux et al., 2009a; Carouge et al., 2010; Pillai et al., 2011, 2012; Broquet et al., 2013) over North American and Eurasian domains in the past decade. However, there still exists lower confidence in estimates over the regions, where there is a lack of both advanced modelling systems at relevant spatio-temporal resolutions and good coverage of ground-based monitoring stations.

In the context of the Indian sub-continent, the inverse-based estimation of fluxes at fine scales is essentially new; hence many questions remain. A number of monitoring sites measuring atmospheric greenhouse gases



have become available in India during the last decade (Tiwari et al., 2011; Lin et al., 2015, Nomura et al., 2021). Aside from the ongoing progress in augmenting observational data streams, it remains challenging to assimilate these data for deducing process-specific information effectively (e.g., McKain et al., 2012; Bréon et al., 2015; Pillai et al., 2016). The limitation of coarse global models in representing observations over the Indian subcontinent is reflected in the analysis made by Patra et al. (2011).

The seasonally reversing South Asian monsoon system is a prominent meteorological phenomenon affecting the Indian subcontinent, which is also expected to influence the terrestrial-atmosphere flux exchanges. Various studies have demonstrated the role of Indian monsoon circulations on regional atmospheric transport by strong south westerly winds during the summer monsoon (June to September) and by north easterly winds during the winter monsoon (October to November) (e.g., Goswami and Xavier, 2005; Krishnamurthy and Shukla, 2007). Monsoon convection transports the boundary layer air into the upper troposphere. Subsequently, air parcels are slowly uplifted by diabatic heating to higher altitudes (e.g., Vogel et al., 2019). An accurate representation of convective vertical transport is very challenging and an important source of uncertainty in current transport models (Willetts et al., 2016). Note that the Asian summer monsoon anticyclone (ASMA) active during the Indian summer monsoon period plays a key role in uplifting trace gases to the upper troposphere and lower stratosphere (e.g., Park et al., 2007). Moreover, a significant component of flux variations can arise from biospheric fluxes (Schimel et al., 2014), which is influenced by variables such as rainfall, availability of radiation, and temperature (Chen et al., 2019). Several studies showed that the monsoon system substantially impacts vegetation growth, generating distinct spatio-temporal patterns of the biogenic fluxes (e.g., Gadgil, 2003; Valsala and Maksyutov, 2013, Ravi Kumar et al., 2016). It is noteworthy that the cropping patterns over India have a strong dependence on seasons and are mainly determined by dry and wet seasons for nearly 65 to 70 % of the country's area except over north-eastern and south-western (Western Ghats) regions of India. In India, wet season crops (Kharif crops cultivated from June to November) including Rice, Millets, and Maize mainly depend on monsoon rain. Dry season crops (Rabi crops, e.g. Wheat, Barley, and Mustard, cultivated from November to April,) are less water-dependent and primarily rely on irrigation (DAC/MA 2015). Therefore, employing a higher resolution modelling over the Indian subcontinent is desirable to better account for fine-scale variations generated by both mesoscale transport processes and surface flux patterns.

This study focuses on accounting for unresolved sub-grid scale variability when employing current generation global models. Assimilation of observations in an inverse framework requires the characterization of these error structures at relevant scales that can be utilized to retrieve source-sink distribution over India. The main objectives of this paper are to describe and quantify the expected spatiotemporal variability of atmospheric $CO_2$ that is not resolved by the current generation global models, quantify to what extent these variations cause uncertainty in flux estimations, and assess how these uncertainties can be minimized by modelling the sub-grid variations in the global models. Specifically, we address the following questions: 1) how good is the level of agreement among global transport models that are used in current generation inversion systems for predicting atmospheric $CO_2$ concentrations over the Indian subcontinent? 2) how large are the variations of atmospheric $CO_2$ that are unresolved by global and regional models, which operate at different spatial scales from 4°× 4° to 0.5°× 0.5° ? 3) what is the role of seasonal changes on generating different patterns in these sub-grid variations of $CO_2$? 4) how much is the uncertainty in the inverse-based flux estimation caused by these unresolved



variations in the coarse models when utilizing a given network of surface observations over the domain? 5) how effectively can we capture the key aspects of the variability and account for it in flux estimations? Information from observations can be better utilized if we improve the atmospheric transport models to resolve the observed

variability as accurately as possible. As a result, the data assimilation system gains significantly (e.g. with increasing weights on observations and performing minimal data filtering), from this for improving the flux estimates.

In this article, we present results based on the analyses of high-resolution simulations at a spatial resolution of 10 km × 10 km for the months of July and November 2017. The year 2017 was characterized by neutral Indian

Ocean Dipole conditions over the Indian Ocean with the beginning of a mild La Nina over the Pacific by the end of the year (NOAA/ESRL, 2022a, b). The month of July represents a monsoon period when the biospheric activity is significant together with atmospheric convection activities. July is also characterized by strong low-pressure system activity over the Bay of Bengal, which results in large rainfall over central India (Krishnamurthy and Ajayamohan, 2010). On the other hand, the month of November is more representative of

post-monsoon wintertime over the Indian subcontinent. We quantify the sub-grid variability using these high-resolution simulations. By designing a pseudo surface observation network over the domain, we investigate the impact of these unresolved variations on the regional flux estimations and assess how a simple parameterization scheme can help in reducing these errors in the global model. To our knowledge, there is no comprehensive published study of this kind over the Indian subcontinent until now assessing the magnitude and impact of

temporal and spatial variability exhibited by atmospheric $CO_2$.

The outline of the paper (see Supplementary Fig. S1) is as follows: Section 2 describes our modelling system, data and methods used for estimating the sub-grid scale variability of $CO_2$. In Sect. 3, we present the global model comparisons and spatial variability analysis, highlighting potential modelling difficulties for estimating the $CO_2$ budget over India. We provide a quantification of the expected sub-grid scale variability based on our

high-resolution simulations, as well as its impact on regional flux estimations. Finally, we discuss the implications of our findings in Sect. 4, suggesting the ways forward to yield an improved estimation of $CO_2$ budgets over India.

## 2 Data & Methodology

We have performed a series of analyses using the simulations generated by our high-resolution modelling system, which is described in Sect. 2.1. Additionally, we have utilized optimized $CO_2$ products at global scales to provide a more comprehensive overview of the typical mismatch between the existing model simulations over the Indian subcontinent at monthly and annual scales (Sect. 2.2). These global model outputs are derived from inverse model simulations, which estimate the source-sink distributions of $CO_2$ and then generate three-

dimensional $CO_2$ concentration fields that are consistent with the optimized posterior fluxes. In this study, the high-resolution simulations are used to quantify the sub-grid scale variability of $CO_2$ that cannot be captured by the global models due to their coarse resolution. For this quantification of the spatial variability, we use the representation error approach described in Sect. 2.3. An observation system simulation experiment (OSSE)


using high-resolution $CO_2$ simulations has been carried out to estimate the impact of the derived sub-grid scale

variations on flux estimations over India via inverse optimization (see Sect. 2.4).

### 2.1 WRF-Chem GHG Modelling System

We use the modelling system WRF-Chem GHG in which the Weather Research and Forecasting model (WRF) version 3.9.1.1 (Skamarock et al., 2008) is coupled with the greenhouse gas module (WRF-Chem-GHG, Beck et al., 2011), implemented as part of the WRF-Chem distribution (WRF-Chem, Grell et al., 2005). For simulating

the atmospheric transport, the model uses fully compressible Eulerian non-hydrostatic equations on Arakawa C-staggered grid, conserving mass, momentum and scalars (Skamarock et al., 2008). In the WRF-Chem GHG (hereafter referred as WRF-GHG), we use the passive tracer chemistry option to simulate changes in $CO_2$ mixing ratios associated with surface fluxes and atmospheric transport. We utilize a biospheric model and emission inventory data to simulate atmospheric $CO_2$ enhancements associated with biogenic and emission

fluxes as described in Sect. 2.1.1 and 2.1.2. Table. 1 summarizes the model configuration, including physics parameterizations and input data used in this study.

The model domain covers a region spanning from 65°E to 100°E and 5°N to 40°N, configured in a Lambert conformal conic (LCC) projection with 307 × 407 grid points. The spatial resolution of the grid is 10 km × 10 km, and the model time-step is 60 s. We have used model output with a temporal resolution of 1 hour for this

study. The simulations are performed using 39 vertical levels with the model top at 50 hPa and 10 levels within the lowest 2 km. WRF-GHG simulations are performed for the entire July and November 2017. Implementation of the WRF-GHG system over the Indian subcontinent enables us to customize it according to the domain features and build a state-of-the-art modelling system, which eventually estimates $CO_2$ fluxes through regional inverse systems. The potential of the WRF-GHG model in simulating fine-scale spatial variability was also

established in previous studies (Ahmadov et al., 2009; Pillai et al., 2011; Park et al., 2018).

### 2.1.1 Representation of biospheric fluxes

We use the Vegetation Photosynthesis and Respiration Model (VPRM) in the modelling system to calculate Net Ecosystem Exchange (NEE) representing the biospheric fluxes (Mahadevan et al., 2008). VPRM is a diagnostic biosphere model, which utilizes remote sensing products: Enhanced Vegetation Index (EVI) and Land Surface

Water Index (LSWI) derived from reflectance data of the Moderate resolution Imaging Spectroradiometer (MODIS) as well as meteorological data: solar radiation and air temperature. In this study, these hourly NEE calculations are performed within WRF-GHG, simultaneously with the meteorology simulations in which NEE is calculated as a sum of gross ecosystem exchange (GEE) and ecosystem respiration ($R_{eco}$). VPRM, in this case, uses the meteorological data provided by WRF-GHG. VPRM uses the SYNMAP vegetation classification

(using the tile approach) (Jung et al., 2006) as well as EVI and LSWI from MODIS surface reflectance data at a resolution of 1 km and 8 days. We aggregate these indices specific for different vegetation types onto the LCC projection for the entire domain at the model's spatial resolution. A number of studies have used VPRM for other regions around the world in which derived NEE shows good prediction skills for hourly to monthly timescales (Ahmadov et al., 2009; Pillai et al., 2011; Liu et al., 2018; Park et al., 2018).

### 2.1.2 Representation of emission fluxes



Anthropogenic $CO_2$ emission fluxes are prescribed from the Emission Database for Global Atmospheric Research (EDGAR) dataset, version 6.0, provided at a horizontal resolution of 0.1° × 0.1° (Crippa et al., 2021). We disaggregate the available annual emission data into hourly emissions using the temporal distribution $CO_2$ profiles (Steinbach et al., 2011; Kretschmer et al., 2014). To represent biomass burning emission, we have used

data from the Global Fire Assimilation System (GFAS) with a spatial resolution of 0.1° × 0.1° and a temporal resolution of one day. GFAS is based on satellite data, which provides the fire emission by assimilating fire radiative power (FRP) observations from MODIS instruments (Kaiser et al., 2012). All these flux data are gridded and projected to WRF-GHG's model domain.

### 2.1.3 Initial and boundary conditions

Meteorological and chemical initial and boundary conditions are required in WRF-GHG to account for the initial state and inflow or background flow. The initial and lateral boundary conditions for the meteorological variables, including horizontal wind components, pressure, specific humidity, sea surface temperature (SST), and the necessary surface initialization fields are obtained from the ERA5 reanalysis dataset of the European Centre for Medium-Range Weather Forecasts (ECMWF), extracted at a horizontal resolution of 25 km and a

temporal resolution of 1 hour (Hersbach et al., 2020). The initial and lateral boundary conditions of $CO_2$ tracers are obtained from the Copernicus Atmosphere Monitoring Service (CAMS, 2.2.4) products (Massart et al., 2016; Agusti-Panareda et al., 2019). We have used the dry air mole fractions of $CO_2$ from the CAMS-GHG, which has a temporal resolution of 6 hour and horizontal resolution of 0.5° × 0.5° with 137 vertical levels. Note that there exists a CAMS product at 9 km × 9 km resolution, which is in the developmental phase and not yet

available to the general public (personal contact: Anna.Agusti-Panareda@ecmwf.int).

We have utilized a simulation strategy to update the initial meteorological conditions for taking advantage of assimilated meteorological fields from ECMWF. The model is reinitialized each day with ECMWF assimilated data at the model starting time of 12.00 UTC (day+0) and runs for 30 hours until 18:00 UTC of the next day (day+1). The first six hours are considered for meteorological spin-up, and the remaining 24 hours (from

day+0,18:00 UTC to day+1, 18:00 UTC) are used for the analysis. The initialization of $CO_2$ is done at the beginning of the first hour of model simulation, which is 00:00 UTC (e.g., Ahmadov et al., 2012; Pillai et al., 2011).

### 2.2 Global model products

We have used optimized products at global scales to examine the differences in the representation of $CO_2$
variability over the Indian subcontinent at monthly and annual scales. Four global inverse modelling products - CarbonTracker, CarboScope, LSCE v18r3 and LSCE FT18r1- available during the year 2017 are used for our analysis (See Table. 2 for more details). The LSCE model version v18r3 (hereafter LSCE) utilizes surface observations for the optimization, and the model version FT18r1 (hereafter LSCE FT) uses satellite retrievals from the Orbiting Carbon Observatory (OCO-2) for the optimization of $CO_2$ fluxes (Chevallier et al., 2005;

Chevallier et al., 2010; Chevallier, 2013). All these above models differ in terms of the model formulations and configuration (e.g., transport and the employed inversion methodology), observational datasets that were assimilated (e.g., data from surface monitoring stations, aircraft missions, ship cruises, AirCore balloon soundings, and satellite's total column retrievals), prior datasets, and spatiotemporal resolutions. None of these





products used ground-based observations from the Indian subcontinent for their optimization, which raises concerns about the reliability of the optimized flux estimations over the region. Hence, it can be assumed that a part of the inter-model differences in predicting the variability is related to the paucity of $CO_2$ observations over the region. To represent the daytime, we have used the concentration fields for the local time ranging from 11:30 to 16:30 from all these models for the analysis.

**2.3 Quantification of spatial variability**

For quantifying the spatial variability due to sub-grid scale processes that cannot be resolved by the coarse resolution models, we follow the approach as described in Pillai et al. (2010). The term 'representation error' indicates the mismatch between the scales of model simulations and observations collected (Pillai et al., 2010; Janjić et al., 2017). In other words, the representation errors arise due to unresolved scales, which could not be captured by the model. Here we calculate the representation errors in the coarse resolution models, which can be

resolved by implementing a high-resolution model at 10 km resolution. It is assumed that the high-resolution simulation captures the majority of the sub-grid scale variability even though it cannot be expected to resolve all observed variability. Most of the current global model simulations are performed at coarse resolutions of several degrees. But with the recent advancement in computational capacity and numerical techniques, a horizontal resolution of 1° × 1° is quite likely achievable for the global data assimilation systems. For estimating the

representation error in a coarse model with a typical spatial resolution of 1° × 1°, we have calculated the standard deviation of $CO_2$ dry air mole fraction simulated by the WRF-GHG model within the coarse grid boxes of 1° × 1° as follows:

$$\sigma_{CO_{2(\text{tot})}} = \sqrt{\frac{1}{n-1}\sum_{j=1}^{n}(m_j - \overline{m})^2}$$  (1)

where $\overline{m} = \frac{1}{n}\sum_{j=1}^{n} m_j$

$n$ is the number of 10 km boxes inside the coarser grid cell of 1° × 1°; $m$ is the $CO_2$ dry air mole fraction corresponding to 10 km boxes; and $\overline{m}$ is the average within the coarser grid cell. So, the estimated values represent the sub-grid scale variability within the coarse model grid cell with a horizontal resolution of 1° × 1°. The representation errors are calculated at corresponding vertical model levels to represent the impact of surface influence and mesoscale transport adequately as predicted by the high-resolution model. As mentioned before,

we assume that the high-resolution simulations represent the realistic distribution of $CO_2$. Further, we assume that the coarse resolution model also has a terrain-following vertical coordinate system and also has the same vertical grid spacing of high-resolution model. As the space-borne instruments can also make the mixing ratio measurements, we extend the analysis to column-averaged dry air mole fraction ($XCO_2$) as measured by the satellite instrument. i.e., $m$ represents either $CO_2$ at a given model level or $XCO_2$. In order to assess the

dependence of representation error on the horizontal resolution of the employed model, we have computed representation error for multiple resolutions ranging from 0.5° × 0.5° to 4° × 4°, in addition to 1° × 1°, which would encompass the resolutions of both present and near-future global inverse modelling systems.

The surface representation errors are calculated using the model simulations from the second model level (mean height is ~200 m from sea level) to avoid the inconsistency that can be generated from inputting emission fluxes



at the first model level. Representation errors are calculated separately for daytime (11:30 to 16:30 local time) and nighttime (23:30 to 4:30 local time) to account for the difference in the sub-grid scale process during these times. The representation error presented in Eq. (1) varies from one model time step to the next. In order to obtain a typical (average) representation error, we compute the monthly average representation error ($\sigma_{CO_2}$) using Eq. (2).

$$\sigma_{CO_2} = \frac{1}{T}\sum_{t=1}^{T}\sigma_{CO_{2(tot)}} \qquad (2)$$

where T is the total number of simulations in a month during daytime or nighttime. Further, we have calculated the representation error ($\sigma_{\overline{CO_2}_{(mon)}}$) using Eq. (3), which only contain systematic component of representation error that can provide important constraints for inversions using both ground-based and satellite observations over India.

$$\sigma_{\overline{CO_2}_{(mon)}} = \sqrt{\frac{1}{n-1}\sum_{j=1}^{n}(M_j - \overline{M})^2} \qquad (3)$$

where $\overline{M} = \frac{1}{n}\sum_{j=1}^{n}M_j$

$n$ is the number of 10 km boxes inside the coarser grid cell of $1° \times 1°$; $M_j$ is the monthly averaged $CO_2$ dry air mole fraction at a 10 km spatial scale; and $\overline{M}$ is the corresponding average within the coarser grid cell of $1°$. The difference between Eq. (1) and Eq. (3) is that we use monthly averaged $CO_2$ concentration values in Eq. (3)
instead of hourly values as in Eq. (1). Both July and November are used to understand the differences in the variability during summer and winter.

Due to the paucity of adequate ground-level observations over India, satellite observations play an essential role in the estimation of $CO_2$ fluxes. Satellite observations can provide column average $CO_2$ ($XCO_2$) concentration with a precision of 1 to 1.5 ppm (O'Dell et al., 2012; Wunch et al., 2017; Liang et al., 2017). In order to utilize
these satellite observations, the transport models being used in the inverse estimation must be highly accurate. Since satellite footprints are smaller ($\sim 2 - 20$ km$^2$) than the current model grid size ($> 100$ km), using these measurements for optimization via inverse modelling introduces spatial representation errors and associated uncertainties in the inferred fluxes. Note that the spatial biases of a few tenths of a ppm in column-averaged $CO_2$ can potentially alter even the annual sub-continental fluxes in the range of tenths of a gigaton of carbon
fluxes (Chevallier et al., 2007, Miller et al., 2007 and Chevallier et al., 2010). To quantify these systematic transport errors when representing satellite measurements in inverse models, we calculate the spatial representation errors for $XCO_2$ that coarse inverse modelling would suffer from using highly precise and accurate satellite measurements.

We have selected monsoon (July) and post-monsoon (November) periods for our analysis to identify the
seasonal changes in the sub-grid variability over India. In July, many low-pressure systems were active in the monsoon trough region (IMD weather reports, https://mausam.imd.gov.in). In general, tropical cyclones in the Asian monsoon region can cause fast uplift of air masses into the upper troposphere and lower stratosphere (e.g.



Li et al., 2021), which may increase the modelling error due to the misrepresentation of the associated mesoscale activity. The presence of enhanced biospheric activity during July can reduce the $CO_2$ concentration

in the lower troposphere. Also, the strong vertical and horizontal mixing due to the monsoon circulation dilutes the $CO_2$ concentration in the atmosphere during July compared to November. The convective activity associated with the Indian summer monsoon was absent during November, however the convection caused by synoptic systems such as tropical cyclones was still present. Such a low-pressure system activity was found over the Bay of Bengal and over the Lakshadweep area ($\approx$ 8° N, 74° E) from 22[nd] November onwards. One of these low-

pressure systems in the Bay of Bengal further developed and intensified as a deep depression and moved to the southeast Arabian Sea and evolved into a severe cyclonic storm (Ockhi) by 30[th] November.

**2.4 Estimation of representation error induced flux uncertainty using pseudo surface measurements**

In order to quantify the impact of representation errors on flux estimations when utilizing surface measurements, we have devised the following strategy. We used nine $CO_2$ surface monitoring sites representing various

geographical regions in India (Fig. 1). Not all these observation stations are currently fully operational or have continuous measurements. We have performed an observation system simulation experiment (OSSE) using high-resolution $CO_2$ simulations generated by the WRF-GHG model for each of these stations. We focus on the biospheric flux component, NEE. The simulated values of coarse models to compare with the observations are obtained from the nine grid cells of the coarse model covering these sites. The pseudo observations for these

sites correspond to the values simulated by the WRF-GHG model at one of the fine grid cells contained in one cell of the coarse model. Since there are 100 fine grid cells per coarse grid cell, 100 different time series are generated and 100 corresponding inversions are performed to obtain robust results. For deducing the contribution of the representation error to the biospheric flux uncertainty, we have taken the following assumptions: 1) the hourly WRF-GHG simulations at 10 km (~ 0.1°) spatial scale represents actual variations in

$CO_2$ mixing ratios of the measurement site, 2) there are no model or observation errors other than representation error, 3) the model captures the spatial and temporal patterns of fluxes correctly, and 4) the contribution from other surface fluxes and background mixing ratio (in ppm) are known. As a first-order simplification for the inversion, we assume that the footprints of each observation site span a radius of 200 km around the site based on our analysis using the Stochastic Time-Inverted Lagrangian Model (STILT, Lin et al., 2003). STILT

footprints indicate that 50% of the sensitivity of a site to fluxes over India is located in a region that has about the same area as a circle with a radius of 200 km. For nine stations, this footprint area covers around 35 % of the total area of India. The STILT is driven with ECMWF IFS (Integrated Forecasting System) meteorological fields and the trajectories are calculated based on 100 virtual particles that are released for each time interval and location. The residence time of particles in the surface layer is weighted by the atmospheric density to

derive the footprints of each location.

In our inversion set-up, we have used the hourly biospheric contribution of the atmospheric $CO_2$ mixing ratios simulated by WRF-GHG over the coarse grid cell of 1° × 1° surrounding the location of each measurement site as OSSE observations ($m_{i,j}(t)$).

$$y_{i,j}(t) \equiv m_{i,j}(t) = \mathbf{H}_{i,j}(t) . \mathbf{F}(\lambda) \qquad (4)$$





where **H** is the transport operator and $\mathbf{F}(\lambda)$ is the flux model in which a subset of parameters $\lambda$ out of total model parameters $p$ will be optimized in the inversion. Here, $i$ ($i$ =1 to 9) represents the nine observation sites and $j$ ($j$ =1 to 100) is the number of WRF-GHG pixels inside the coarser grid cell of 1° × 1°..

The modelled biospheric $CO_2$ signal ($\overline{m}_i$) for the inversion is given by:

$$\overline{m}_i(t) = m_{i,j}(t) + \varepsilon_{i,j}(t) \qquad (5)$$

The modelled values deviate from the observations by a representation error $\varepsilon_{i,j}(t)$. Since the modelled values ($\overline{m}_i$) correspond to the mean of the 100 fine grid cells, the simulated values at site $i$ are given as:

$$\overline{m}_i(t) = \frac{1}{100}\sum_1^{100} m_{i,j}(t) \qquad (6)$$

Here, $\mathbf{F}(\lambda)$ is taken as linearly dependent on $\lambda$ ; hence can be expressed as

$$\mathbf{F}(\lambda) = \mathbf{\Phi}.\lambda \qquad (7)$$

where $\mathbf{\Phi}$ is the biospheric flux (NEE) distribution over the region.

In the inversion, we retrieve monthly NEE by utilizing hourly $m_{i,j}(t)$ and $\overline{m}_i(t)$ over a month. For OSSE and uncertainty flux estimation, we use the VPRM-derived NEE fluxes as the "true" fluxes (see Sect. 2.1.1). By this inverse modelling design, we require to perform 100 inversions per site, each of which uses a realization of

representation error to estimate the corresponding realization of the resulting uncertainty in the retrieved fluxes.

Both the observation and simulation vector have 6480 (=9×30×24) elements for a month having 30 days, and the state vector has 9 elements corresponding to scaling factors of fluxes for that month over regions around the 9 sites (see Fig. 1). In other words, each site has been assigned with one scaling factor for NEE, and there is a total of 9 scaling factors for a given month. We use a unit vector $\lambda$ as prior scaling factors. The prior uncertainty

is neglected here, as the expected impact of the representation error on the retrieved fluxes is significantly smaller than typical prior uncertainties assumed in Bayesian inversions (on the order of 50% – 100% for biospheric fluxes). Hence neglecting this prior uncertainty does not have a large impact on our results. The inversion retrieves optimized scaling factors $\lambda_{retr}$.

We have performed 100 inversions per site, and the scaling factors are retrieved by minimising the cost function

for each observation station:

$$J(\lambda_{i,j}) = \frac{1}{T}\sum_{t=1}^{T}(m_{i,j}(t) - \overline{m}_i(t)\lambda_{i,j})^2 \qquad (8)$$

where T is the number of observations for a month. Minimizing these cost functions results in an optimized estimate of scaling factors $\lambda_{retr}$, which is a vector of scaling factors with nine elements ($\lambda_{retr,i}$) for each of the 100 inversion cases.

By this inverse design, the deviation of posterior fluxes from the true fluxes over India is thus the uncertainty in retrieved fluxes, $\mathbf{S}_{rep}$ , that arises solely due to the contribution from the representation error. Standard deviation





of the scaling factors from these 100 inversions ($\sigma_{\lambda_{retr}}$ ) are used to retrieve flux uncertainty. $\mathbf{S}_{rep}$ is obtained as follows:

$$S_{rep} = \sqrt{\sum_{k=1}^{K}\left(S_{\lambda_{retr,k}}\Phi_{true,k}\right)^2} \qquad (9)$$

where $\Phi_{true}$ is the monthly VPRM biospheric flux (NEE) over the Indian region and $k$ is the number of pixels (33141 pixels) over the Indian region. Here, $\mathbf{S}_{\lambda_{retr}}$ has the dimension of Indian region at a 10 km spatial resolution and is defined in such a way that all the grids (at 10 km spatial resolution) other than the grids within the influence region (200 km radius around the station) of each station is given with zero values (21335 pixels) and the grids in the influence region of each station (11806 pixels) is given with the corresponding values of

$\sigma_{\lambda_{retr,i}}$. This way, the approach doesn't depend on Eq. (1) to Eq. (3), but shows the impact of difference between $m_j$ and $\overline{m}$ on retrieved fluxes.

Any temporal correlations in the representation error are not considered for this experiment. We have performed the inversion separately for daytime and nighttime values to identify the impact of diurnal variations of representation errors on flux uncertainty. Note that by following the above inversion design and assumptions,

there is a high likelihood of underestimating the impact of the modelling error on flux estimations since we have not considered other sources of uncertainties such as model transport uncertainty and inappropriate prior assumptions. Thus, the quantification of flux uncertainty using this approach can be inferred as the lower bound of the uncertainty (i.e., the minimum flux uncertainty one may expect while estimating fluxes using a model with a grid cell of 1°× 1° and 9 stations with the representativeness of 200 km).

**3 Results and Discussions**

**3.1 Agreement among global models**

We first analyse the level of agreement among current-generation global transport models in simulating $CO_2$ concentration over the Indian subcontinent. Note that a mere agreement among the coarse models is not sufficient to justify the models' performance over the region due to their plausibly large model errors in

common and interdependency in terms of data sources. We restrict this analysis to daytime-only values since different processes control the variability of $CO_2$ concentration at daytime and nighttime, and simulating nighttime variability is more complicated than the daytime (Lauvaux et al., 2009a). For a consistent comparison among global models, all the products are sampled at the same time for the region spanning from 67° E to 98° E and 7° N to 38° N. Figure 2a depicts the annual vertical profiles of $CO_2$ concentration, showing models'

discrepancy in simulating the vertical gradients in concentration values including the boundary layer and the free troposphere. A notable difference is observed in the simulation of the gradient within the boundary layer. The magnitude and the height up to which this positive gradient is observed are different for these models. LSCE (both versions) has the largest positive gradient among these models (~1ppm). It shows the maximum concentration at around 700 m height and then a decrease in concentration. CarbonTracker also shows this

positive gradient in the surface layers up to a height of 900 m. But the gradient is much smaller compared to the other two models. Among these four models, CarboScope does not exhibit this tendency in the lower atmosphere. Its concentration decreases linearly from the surface as the height increases.



The seasonal variability of $CO_2$ uptake through photosynthesis, release through ecosystem respiration, and vertical transport is seen while analysing the monthly averaged $CO_2$ concentration profiles over the Indian subcontinent (Figs. 2b and 3). Comparatively lower surface $CO_2$ concentrations are found during months with an active biosphere (June to October) than the rest of the period, owing to the higher ecosystem productivity over the northern hemisphere and particularly over the Indian subcontinent in response to the availability of monsoon rainfall. Also, the presence of strong southwest monsoon winds from June to September may result in bringing $CO_2$ depleted air from the southern hemisphere and thereby lowering the $CO_2$ concentration over the domain. While comparing the seasonal maximum (May) and minimum (September) of $CO_2$ concentrations measured at the Mauna Loa observatory (MLO) located in Hawaii, Fig. 2b shows a temporal shift of around one month for exhibiting seasonal maximum (April) and minimum (August) $CO_2$ concentrations. This temporal shift is attributed to the differential impacts of anthropogenic and terrestrial ecosystem activities on atmospheric concentration as well as the long-distance transfer of atmospheric carbon dioxide to remote location (Nomura et al., 2021). MLO observations are generally representative of global mean $CO_2$ due to the minimal influence of terrestrial ecosystems and anthropogenic activities at remote location. The seasonal variation of monthly averaged $CO_2$ seen over the Indian subcontinent is mostly dominated by terrestrial carbon fluxes, i.e., net ecosystem exchange (NEE) as seen from the VPRM simulations (see Supplementary Fig. S2).

Further, we see a $CO_2$ vertical profile with a small vertical gradient (~0.5 ppm within an altitude range of ~500 m to 4000 m) from June to October (Fig. 3). This is likely linked to the increased convective activities associated with the monsoon. The strong vertical gradient in the surface levels as simulated by the LSCE model during the monsoon period is little plausible given the strong vertical mixing expected for this convective period. The considerable inter-model variation in monthly averaged $CO_2$ concentration profiles as predicted by different global models is problematic as it indicates significant uncertainties in flux estimations over India. A part of this discrepancy can come from the coarse resolution global model's inability to represent transport processes like convection and vertical mixing, strength and distribution of anthropogenic sources and ecosystem activities that operate at fine scales. The extent of this unresolved variability in global models is further explored in Sect. 3.2. The spatial distribution of $CO_2$ concentration shows structural differences among these models (see Supplementary Fig. S3), indicating a substantial knowledge gap in representing atmospheric $CO_2$ variability over the Indian subcontinent, which can have severe implications for the country's carbon budget estimations.

### 3.2 Representation errors in global transport models

The spatio-temporal variability of representation error and the influence of various factors in creating this variability are examined here. The larger the variations that are caused due to sub-grid processes within the grid box of 1° × 1°, the larger the representation error. The derived seasonal differences in structural patterns of the sub-grid variability facilitate to 1) quantify what would be typical representation errors associated with incorporating seasonally varying observations into atmospheric models 2) determine what drives the seasonality in sub-grid variability and ultimately 3) design a possible parameterization of representation error with a seasonal component in the inverse modelling framework as well as identify periods or seasons where the use of this parameterization would be valid to improve our estimations of $CO_2$ fluxes. Further, the seasonal spatial variability analysis of column averages can provide useful information for the satellite community to gap-fill the





satellite soundings over India when large data gaps and low sounding precision on daily or monthly time scales are present, especially the case for monsoon periods in India.

### 3.2.1 Spatio-temporal patterns

Representation errors in the surface $CO_2$ concentrations of a global model at a spatial resolution of $1° \times 1°$ for

July and November are shown in Fig. 4. The representation error at $1° \times 1°$ spatial scale reaches values ranging from 0.5 ppm to 5 ppm, which are comparable to the magnitude of variability at hotspot emission regions or half of the seasonal variability of $CO_2$ over the region (see Fig. 2b). The median representation error is 1.2 ppm at the surface, which is considerably larger than the measurement errors. In the case of high accuracy in situ measurements, the typical uncertainty for $CO_2$ measurements is less than 0.1 ppm (Andrews et al., 2014). A

remarkable feature is the presence of very high representation error over North-East and Western Ghats regions, where the biosphere activity is very prominent. The heterogeneous distribution of biosphere fluxes generates significant sub-grid scale variability that leads to high representation error. Also, we can find high representation error along the foothills of the Himalayas. In addition to the complex terrain, the region over the Ganges basin is characterized by increased anthropogenic activity, which contributes to a larger representation error surrounding

this region. High representation error is also found in the coastal regions, ranging from 2 ppm to 5 ppm (median of 4 ppm) due to the temporal covariance between the coastal meteorology and exchange fluxes. The $CO_2$ fluxes from coastal regions can be transported over the ocean and accumulated in the shallow boundary layer over the ocean. The shallow boundary layer is a characteristic of the marine atmosphere due to the less vertical mixing compared to land regions. Horizontal $CO_2$ gradients can also be generated by the influence of highly varying

biospheric fluxes under different advection patterns over the land and ocean boundary. A similar mechanism is applicable to mountain regions where temporal covariance of mountain-valley circulation and respired $CO_2$ fluxes are regulated by atmospheric radiation. The terrain-following coordinates as used in the model may also result in spurious tracer concentration gradients over the steep mountain terrain (Beck et al., 2020; Skamarock et al., 2021; Park et al., 2019). Though the mesoscale models are expected to perform better in simulating $CO_2$

variations over the complex terrain than the coarse models (e.g. Engelen et al., 2002; Gerbig et al., 2003; Ahmadov et al., 2007; Corbin et al., 2008; Lauvaux et al., 2009b; Pillai et al., 2011; Uebel et al., 2017; Agustí-Panareda et al., 2019), they may also suffer from the inadequate representation of complex weather features and associated variability. We can also find individual cells with high representation errors associated with point emission sources such as cities, mining sites, and coal-fired power plants at different parts of the domain. The

daily variations in surface representation errors are small within a month, although there exists a clear distinction between daytime and nighttime values (Figure not shown). The nighttime representation error is higher (e.g. a median value of 1.5 ppm for surface during November) compared to the daytime representation error (e.g. a median value of 1.1 ppm for surface during November) throughout the analysed domain. This is expected due to the coupling between nocturnal shallow transport and different flux processes accentuating local

effects. During the nighttime, photosynthesis is absent, and respiration is the major biospheric activity, leading to an increase in $CO_2$ concentration in the atmosphere. The large heterogeneity in flux distribution that is mostly from respired $CO_2$ fluxes, the shallow boundary layer processes and the weak nocturnal turbulence cause $CO_2$ to be accumulated locally near the surface with large variations. Compared to July, we find higher representation


error in November owing to the wintertime transport with decreased vertical mixing and heterogeneous biospheric uptake (see Fig. 4).

In the case of $XCO_2$, the magnitude of sub-grid scale variability is much smaller than that of surface $CO_2$ (Fig. 5), but it follows a similar spatial pattern. This confirms the dominance of surface-level processes in causing sub-grid variability of column averages. The sub-grid scale variability in $XCO_2$ reaches up to 2 ppm in some parts of the region, especially where there are high variations in topographic features or point emission sources.

The estimated column representation error is thus capable of causing significant biases in the satellite inferred $CO_2$ fluxes over these regions. Also, the representation error for a large part of the domain is found to be above 0.5 ppm, which is around half of the typical precision of current satellite measurements. Note that the representation error reported here is different from satellite measurement errors (e.g. spectroscopic retrieval error or sampling biases) and tends to be systematic in nature.

Figure 6 shows the statistical distribution of the representation error ($\sigma_{CO_2}$) sampled over India, during July and November, separated by daytime and nighttime. July shows a median surface representation error of 0.9 ppm and 1.1 ppm during daytime and nighttime respectively, while November shows a median value of 1.1 ppm and 1.4 ppm for daytime and nighttime respectively. In July, 95 % of the representation error is less than 2.1 ppm for daytime (3.9 ppm for nighttime) while it is 3 ppm for daytime (4.2 ppm for nighttime) for November. For

column average, median values for representation error are 0.3 ppm and 0.4 ppm for July daytime and November daytime respectively.

To further reduce the effect of random error that might be introduced by short-term weather phenomena, the representation errors ($\sigma_{\overline{CO_2}_{(mon)}}$) are calculated from the monthly averaged $CO_2$ field and are denoted as a systematic error (Fig. 6). Uncorrelated errors are expected to decrease when averaging over a sufficiently long

period. As expected, the median values of the systematic representation errors are smaller for all cases, showing the effect of random errors. Especially for November when the cyclonic event was present, the values of the systematic errors (in the 95% percentile) for the surface $CO_2$ are considerably lower than total errors, reducing from 3 ppm (daytime) and 4.2 ppm (nighttime) to 2.2 ppm (daytime) and 3 ppm (nighttime). In the case of column $CO_2$, this reduction is from 1.1 ppm (daytime) and 0.9 ppm (nighttime) to 0.8 ppm (daytime) and 0.7

ppm (nighttime) in the 95% percentile. In contrast to surface representation error (Fig. 6a), median values of nighttime representation errors are found to be slightly lower than daytime representation error for column average (Fig. 6b). To assess the dependence of representation error on possible horizontal resolutions of the global models, we have further derived the representation errors for different spatial resolutions between 0.5° and 4°. As expected, we see reductions in representation errors for both surface and column averaged $CO_2$ with

increasing horizontal resolution of the model (See Fig. 7 & supplementary Fig. S4). During July, the median surface representation error reduced from 1.6 ppm (2 ppm) to 0.6 ppm (0.7 ppm) during daytime (nighttime) while increasing horizontal resolution from 4° to 0.5°. This increment in spatial resolution has also resulted in similar error reductions in November during which the median of surface representation error shows a reduction from 2.4 ppm (2.8 ppm) to 0.7 ppm (0.9 ppm) during daytime (nighttime).  In the case of column-averaged

values, the median representation error decreased from 0.7 ppm (0.6 ppm) to 0.25 ppm (0.2 ppm) during July daytime (nighttime) and from 0.95 ppm (0.9 ppm) to 0.25 ppm (0.2 ppm) during November daytime



(nighttime). The spatial distribution of representation errors for a model with a horizontal grid resolution of 0.5°× 0.5° (e.g. regional models) is provided in Supplementary Figs. S5 and S6. On average, we find ~33 to 36 % of decrease in daytime representation errors for both months when increasing model grid resolution from 1° to 0.5°. There exists a similar spatial pattern of representation errors for both resolutions of 0.5° and 1°. Though our results indicate a reduction of representation error for regional models with a typical resolution of 0.5° compared to global models with 1° spatial resolution, the emission hotspots and point sources are still pronounced with high sub-grid scale variability, especially during nighttime. The above analyses indicate that the sub-grid variability alone can produce significantly higher errors compared to the measurement errors (e.g., 0.1 ppm as per WMO standards for surface measurements), which necessitates a proper treatment of these errors in models for the optimal estimation of $CO_2$ fluxes.

### 3.2.2 Vertical distribution

Figure 8 shows the vertical profile of representation error distribution within different altitude bins. We find that the maximum representation error is in the surface layer, and most of the higher values are found to be within the lowest 4-6 km bins. Also, sub-grid scale variability decreases sharply with increasing altitude. This dominance of variability in surface concentration can be explained by surface flux heterogeneity influencing mole fractions in lower atmospheric layers (PBL) as described in van der Molen and Dolman (2007) and Pillai et al. (2010). There is a slight increase in representation error in the upper tropospheric levels near 12 to 14 km altitude range. This may be associated with the presence of strong circulations in the upper troposphere and lower stratosphere, such as subtropical westerly jets.

### 3.3. Influence of terrain heterogeneity and flux variability on representation errors

Here we explore the factors influencing the size and patterns of the representation error in coarse models. For this, statistical relationships between representation error and possible explanatory variables are examined for both surface and column-averaged $CO_2$. Identifying these factors influencing representation errors and quantifying their local effects facilitate us to further investigate on how these biases in retrieved fluxes can be minimized in global models (see Sect. 3.5).

We find a significant influence of terrain heterogeneity on representation error, which is evident from the spatial maps in Figs. 4 and 5, where the largest sub-grid scale variations are found in the Himalayan regions. Spatial variations in topography produce mesoscale circulation patterns and corresponding variations in atmospheric $CO_2$ at fine scales. At the same time, there is a plausible additional error in global model simulations related to the insufficient resolution of vertical grids necessary to account for different surface influences (e.g. mountain vs valley). This effect of coarse vertical resolution is excluded in our representation error estimates by preserving the vertical grids used for the high-resolution simulations. To further explore the importance of using the high-resolution topography data on representing the $CO_2$ variability, we analyse the dependence of terrain variations (as derived from the standard deviation of terrain height) on the distribution of the representation error. We have estimated the statistical dependence of topographic variability within the global climate models' grids on corresponding representation error to estimate the relation between them. Topographic variability within 1° × 1° spatial box is estimated as the standard deviation of topography (m) for all 10 km × 10 km boxes within the larger grid, and is denoted as $\sigma_{topo}$. Bins are created based on the values of this topographic



variability, in which different points from different parts of the domain are binned together on the basis of their standard deviation of topography. Each bin is created with a size of 50 m variation in terrain height. The linear fit is estimated between the average value of topographic variability within a bin and the average value of representation error of the corresponding points in the bin. Our results show that the terrain heterogeneity alone can explain about 20-48% of the surface representation errors over the domain. In a similar way, we have

estimated the influence of topographic variability on representation error in the column-averaged model simulations. It is found that topography alone can explain 45-52 % of representation errors in the column-averaged simulations.

Further, we estimate the statistical relationship between the surface flux heterogeneity and representation error. The surface representation error is strongly linked to the biosphere flux variability, and the relationship between

heterogeneity in biospheric surface flux (as derived from the standard deviation of VPRM-derived NEE fluxes, denoted as $\sigma_{bio}$) and representation errors depends on the time of the day and season. During daytime when there is strong ecosystem activity, the dependence of representation error ($\sigma_{CO_2}$) on $\sigma_{bio}$ of surface and column $CO_2$ is found to be ~75-80 % and ~66-74 % respectively. $\sigma_{bio}$ explains about 62% for the surface $CO_2$ variability and 48 % for the column variability during July nighttime. However, $\sigma_{CO_2}$ and $\sigma_{bio}$ are less correlated

(23 % for surface and 19 % for column) during November nighttime. The diurnal difference in the dependence of $\sigma_{bio}$ on representation error can be explained by the increased magnitude and spatial variability of daytime biospheric fluxes in the growing season (primarily due to photosynthesis activities) compared to nighttime fluxes. Moreover, poor vertical mixing under the stable nocturnal atmospheric conditions with more advection and drainage flow reduces the influence of surface fluxes on spatial variability of mixing ratios. The dependence

of representation error on the anthropogenic flux heterogeneity (as derived from the standard deviation of EDGAR fluxes, denoted as $\sigma_{ant}$) is found to be negligible except for nighttime (13–30 %). We find less influence of seasonality on the relationship between anthropogenic surface flux heterogeneity and representation errors (see Supplementary Table S1). Similar to the above analysis with $\sigma_{bio}$, the combined effect of atmospheric stability and flux heterogeneity can explain the diurnal differences of the relationship between $\sigma_{ant}$

and $\sigma_{CO_2}$.

In case of the variability of monthly averages, we see that $\sigma_{\overline{CO_2}(mon)}$ is well explained by $\sigma_{bio}$ during daytime (see Supplementary Table S2), as expected. A similar strong correlation can be seen between $\sigma_{\overline{CO_2}(mon)}$ and $\sigma_{bio}$ (23–69 %) during nighttime for surface variability of $CO_2$, while there exists only less dependence of nocturnal column $CO_2$ variability on local fluxes. This shows the decoupling of the mixing ratios in other parts

of the column from the surface during the night due to less vertical mixing, combined with more drainage flow in the nocturnal boundary layer, which reduces the effect of surface flux variability on the column $CO_2$ variability.



In general, the above analysis underlines the need for using accurate Digital Elevation Models (DEMs) in the atmospheric transport models as one of the most critical datasets for determining the mesoscale atmospheric
flows adequately. Further, the results also indicate the importance of utilising surface fluxes at high resolution.

### 3.4 Estimation of NEE flux uncertainty due to representation error

By following the assumptions and approach as given in Sect. 2.4, we have estimated the NEE flux uncertainty resulting from the representation errors. The results based on the OSSEs for nine observation sites are given in Table. 3. The scaling factors, which are calculated separately for each site by adjusting the prior fluxes using
pseudo-observations, are applied to the VPRM monthly fluxes. The total NEE flux for India estimated by VPRM for July and November are -373.3 $MtCO_2$ per month and -417.1 $MtCO_2$ per month, respectively. The flux uncertainties over India that arise solely due to the contribution from the representation error are estimated to be 38.59 (daytime observations) to 30.14 (nighttime observations) $MtCO_2$ per month (10.33% to 8.07%) for July and 18.42 (daytime observations) to 13.34 (nighttime observations) $MtCO_2$ per month (4.4% to 3.1%) for
November while utilizing data from nine observation stations. The maximum flux uncertainty was found for July due to the enhanced biosphere activity and unresolved convection activities. The estimated uncertainties are considerable for the carbon budget assessment especially given that these errors arise solely from the global models' representation error. Note that calculated representation error does not include other transport error sources such as advection, convection or vertical mixing.

### 3.5 Possible treatment of representation error in the global model

The simplest possible way to minimize the uncertainty in flux estimation using a coarse model is to construct a parameterization model that can account for the representation error using explanatory variables. For this, we create a multivariate model to capture spatial patterns in the representation error. Employing this parameterization in a global model would thus redefine the likelihood of better estimates (improving the state of
knowledge) with variance greater than that of the measurement error in the inverse framework by minimizing the modelling error. The multivariate linear model with explanatory variables that include sub-grid variations of terrain ($\sigma_{topo}$), biospheric ($\sigma_{bio}$) and anthropogenic ($\sigma_{ant}$) fluxes remarkably captures the derived column representation error all over the Indian region during July daytime with a $R^2$ value of 0.96 (Fig. 9). The difference between the modelled and derived representation error is found to be well below 0.5 ppm in most parts of the domain. Similarly, we have modelled the surface representation error using the linear model with
these three explanatory variables and found that the proposed model could capture the derived surface representation error well with a deviation less than 1 ppm in most of the regions (see Supplementary Fig. S7 and Supplementary Table S1 and S2). More work is needed to demonstrate the extent of applicability of this method to minimize the flux uncertainties while utilizing actual observations. Nevertheless, the above finding provides a
possibility for a parameterization that can be further developed in inverse models or data assimilation systems, which defines the degrees of freedom for describing the posterior states. Applying this parameterization scheme to the specific problem requires a high-resolution map of the terrain and prior information on anthropogenic and biogenic fluxes. The uncertainties in the topography can significantly impact flux estimation, and the likely reduction of flux uncertainty depends on the accuracy of the DEM employed. The caveat of this linear model is
that the uncorrelated spatial variability in the prior and true states of the fluxes is ignored in the present form,



which cannot be the case for the real inverse problems. This assumption obviously hampers the system in achieving the maximum reduction in uncertainty, and further study is needed to refine this model from a practical perspective. We emphasise, however, that the above parameterization does not require a high-resolution simulation of transport, which has high computational costs.

### 4. Conclusion

Given the upcoming availability of atmospheric observations over India, significant effort is required to critically enhance the modelling capabilities to derive carbon budgets over India within the definite confidence intervals and at scales relevant to the ecosystem and countrywide policy-making. The misrepresentation of mesoscale transport phenomena and unresolved flux variations in modelling systems operating on coarse grids hinders the optimal utilization of observations. In this context, the present study quantifies the spatial variability of atmospheric $CO_2$ mixing ratio over India that is not resolved by the coarse models and assesses their impact on flux estimations. We demonstrate the potential of a simple parameterization scheme to model these unresolved variations in the coarse models for minimizing the uncertainty in retrieved fluxes.

A large spread among global model simulations in representing monthly averaged $CO_2$ concentration profiles indicates a considerable knowledge gap in the estimations of fluxes even at a monthly scale. It can be argued that a significant part of these differences arises due to the lack of observational constraints over India, which leads to a possible compensatory model artefact over this region in order to match the global mass constraint. At the same time, it is also expected that the spatial variability of the observed atmospheric $CO_2$ mole fractions can be large so that these coarse models fail to represent them adequately. For instance, we find that the unresolved variations (representation error) of global models with a spatial resolution of 1° × 1° can be ~1.5 ppm on average for the surface $CO_2$ that is even larger than the currently reported inter-global model differences. Similarly, the average representation error estimated for the column-averaged $CO_2$ is ~1.1 ppm. These estimated values are larger than the corresponding measurement errors, which cause the inverse optimization to infer a state that is not close to the truth as is required in the regional $CO_2$ budget for various applications.

Coastal areas and mountains have particularly high representation errors (≈4 ppm for surface $CO_2$). Emission hotspots can also lead to significant $CO_2$ variability near the surface as large as ≈8 ppm. Larger values are typically associated with the nocturnal shallow boundary layer dynamics and the stronger respiration signal with considerable flux variability. These findings are consistent with Pillai et al. (2010), which show that there exist spatial differences in the sub-grid variability for both surface and column $CO_2$. Although the magnitude of the sub-grid variability of the total column is an order of magnitude smaller than the variability at the surface, the spatial pattern remains similar for both, owing to the dominance of surface heterogeneity in topography and fluxes. With the underlying assumptions, the total uncertainty in optimized fluxes solely due to the unresolved sub-grid variations is estimated to be 3.1 to 10.3% of the total NEE while utilizing pseudo-data from nine observation stations over India. Increasing the spatial and temporal resolutions of the transport models can generally capture the mesoscale features and associated $CO_2$ gradients, thereby reducing the representation error. Increasing the model's resolution from 1° to 0.5° has shown an improvement in capturing variability with representation error reduction of 33% and 36% for summer time and winter time, respectively. By showing the existence of unresolved variability in 0.5° resolution with a similar spatial pattern of error as of 1° spatial





resolution, we demonstrate the need for a much finer resolution than 0.5° for representing the atmospheric $CO_2$ variability over India. However, merely increasing the resolution without having a realistic representation of terrain heterogeneity and flux (both natural and anthropogenic) variability would not be beneficial. The uncertainties in the high-resolution fluxes can worsen the model's skills, whose effect would not be more pronounced at coarser resolutions due to the diffusive nature of fluxes, as seen in Agustí-Panareda et al. (2019).

A parameterization scheme with explanatory variables of sub-grid variations of terrain, biospheric and anthropogenic fluxes is shown to capture a considerable fraction of expected representation error in the global model. The proposed method is easy to implement in the coarse models as it does not require computationally expensive transport simulations at high resolution. As we see a significant dependence of the distribution of sub-grid variability on terrain variations, our results reinforce the requirement for using accurate DEMs in the atmospheric transport models. The biosphere flux variability explains about 62 to 80% of the surface

representation errors over the Indian region, indicating the need for using precise high-resolution surface fluxes.

Overall, we show that the mesoscale transport mechanisms and flux variability contribute to fine-scale $CO_2$ variations that the current-generation models cannot resolve. Our findings indicate that the models need to be critically improved to capture mesoscale variations associated with horizontal and vertical transport and fine-scale flux variability to maximize the potential of highly precise and accurate measurements. Our results provide

a baseline for overcoming the shortcomings mentioned above and accounting for the realistic distribution of atmospheric $CO_2$ to improve the estimation of surface fluxes through inverse modelling.

**Code/Data availability**

The WRF-Chem source code is publicly available at https://ruc.noaa.gov/wrf/wrf-chem/ (last access: 10 August

2019). The CarbonTracker (CT-2019B) products are available online at http://carbontracker.noaa.gov (last access: 21 July 2020, Jacobson et al., 2020). The data from the CarboScope inversion system are available online at http://www.bgc-jena.mpg.de/CarboScope/ (last access: 20 July 2020, Rödenbeck et al., 2003). The data from the LSCE modelling system used in this study are available at http://atmosphere.copernicus.eu (last access: 22 July 2020, Chevallier et al., 2019). The sub-grid variability products based on the WRF-GHG model

simulations can be accessed from https://zenodo.org/record/6616466 (last access: 23 May 2022, Thilakan and Pillai, 2022). The WRF-GHG model $CO_2$ simulations used for this study are available upon request to the corresponding author, Dhanyalekshmi Pillai (dhanya@iiserb.ac.in, kdhanya@bgc-jena.mpg.de). The EDGAR data used in this study are publicly available at https://edgar.jrc.ec.europa.eu/ (last access: 15 March 2020, Crippa et al., 2018). The GFAS data are publicly available at http://apps.ecmwf.int/datasets/data/cams-gfas/ (last

access: 15 March 2020, Kaiser et al., 2012). The ERA5 data are available at https://cds.climate.copernicus.eu/cdsapp#!/home (last access: 18 March 2020, Hersbach et al., (2020)).

**Author Contribution**

DP designed the study and performed the model simulations. VT performed the analysis and wrote the paper.

VT and DP interpreted the results. CG, MG, AR and TAM provided significant input to the interpretation, and the improvement of the paper. All authors discussed the results and commented on the paper.



**Competing interests**

The authors declare they have no conflict of interest.

**Acknowledgements**

This study is supported by the funding from the Max Planck Society allocated to the Max Planck Partner Group at IISERB and the Science and Engineering Research Board (SERB) through Early Career Research Award (ECR/2018/001111) to DP. TAM acknowledges the financial support provided by SERB grant (Junior Research Fellowship). We acknowledge the support of IISERB's high performance cluster system for computations, data analysis and visualisation. The WRF-Chem simulations were done on the high-performance cluster Mistral of

the Deutsches Klimarechenzentrum GmbH (DKRZ). We thank the Editor and both referees for their constructive comments.

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

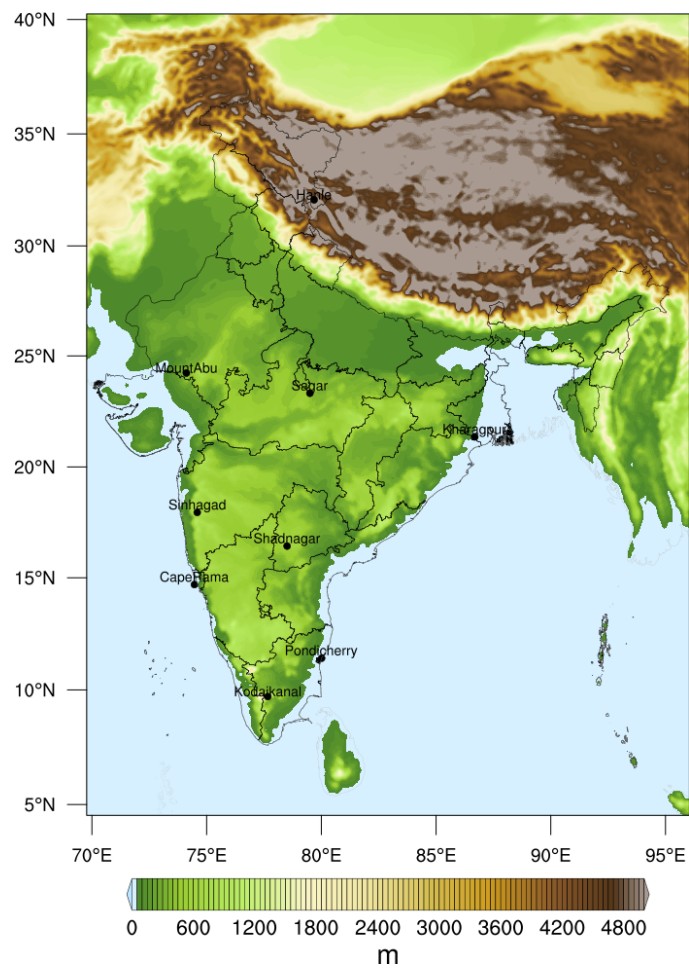

**Figure 1: The WRF-GHG model domain used in this study, showing topography. The CO$_2$ monitoring sites over India used for the OSSE experiments are marked. Not all these observation stations are currently fully operational. The colour scale is restricted to 5000 m for the better visualization of terrain details over the Indian subcontinent.**


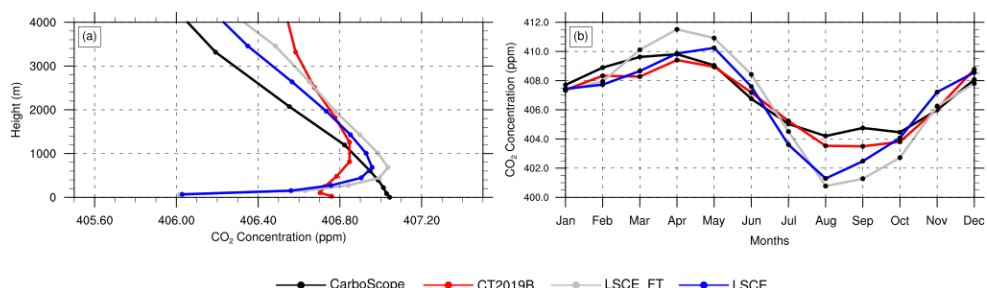


**Figure 2: Comparison of global models over the model domain during daytime (11:30 to 16:30 local time) in 2017. a) Annually averaged vertical profiles of $CO_2$ concentration in the lower troposphere b) Time series of monthly averaged $CO_2$ concentration at surface (~100 m above surface).**

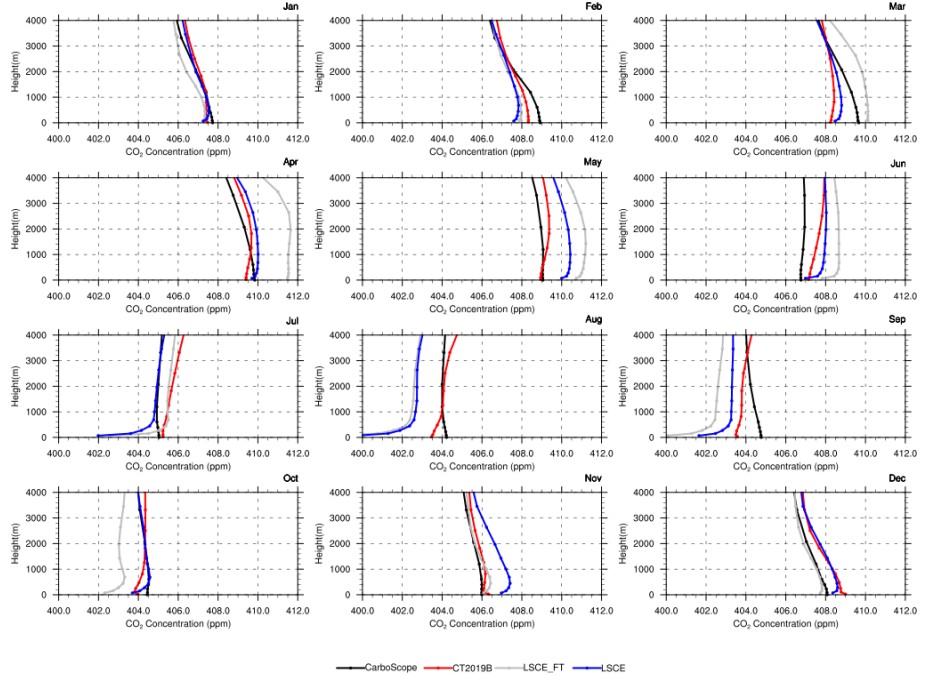

**Figure 3: Comparison of average monthly vertical profiles of $CO_2$ concentration from global atmospheric transport models over the model domain during daytime (11:30 to 16:30 local time) in 2017. Panels show data for respective months as indicated on the top of each panel.**



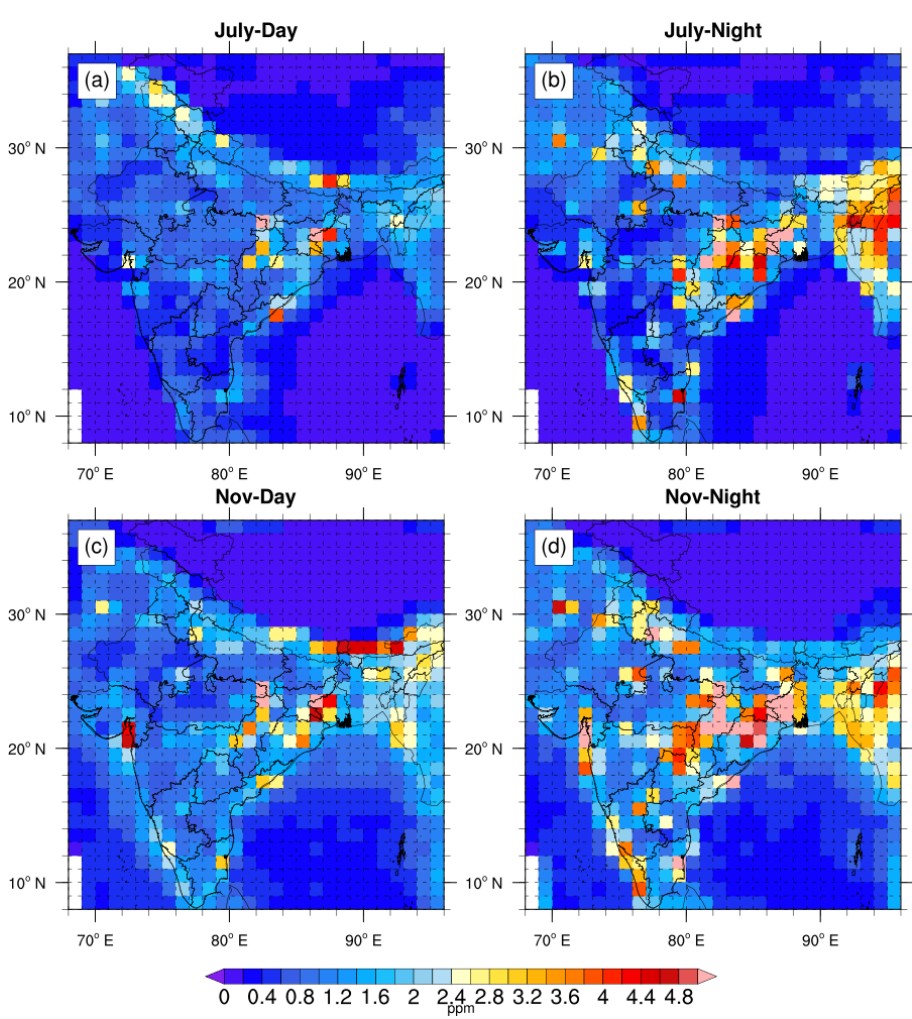


**Figure 4: Monthly averaged values of representation error estimated for surface CO$_2$ concentration (second model level, mean height is ~200 m from sea level) over the region 8° N to 37° N and 68° E to 96° E during 2017. a) July daytime (11:30 to 16:30 local time) b) July nighttime (23:30 to 4:30 local time). c) November daytime. d) November nighttime.**



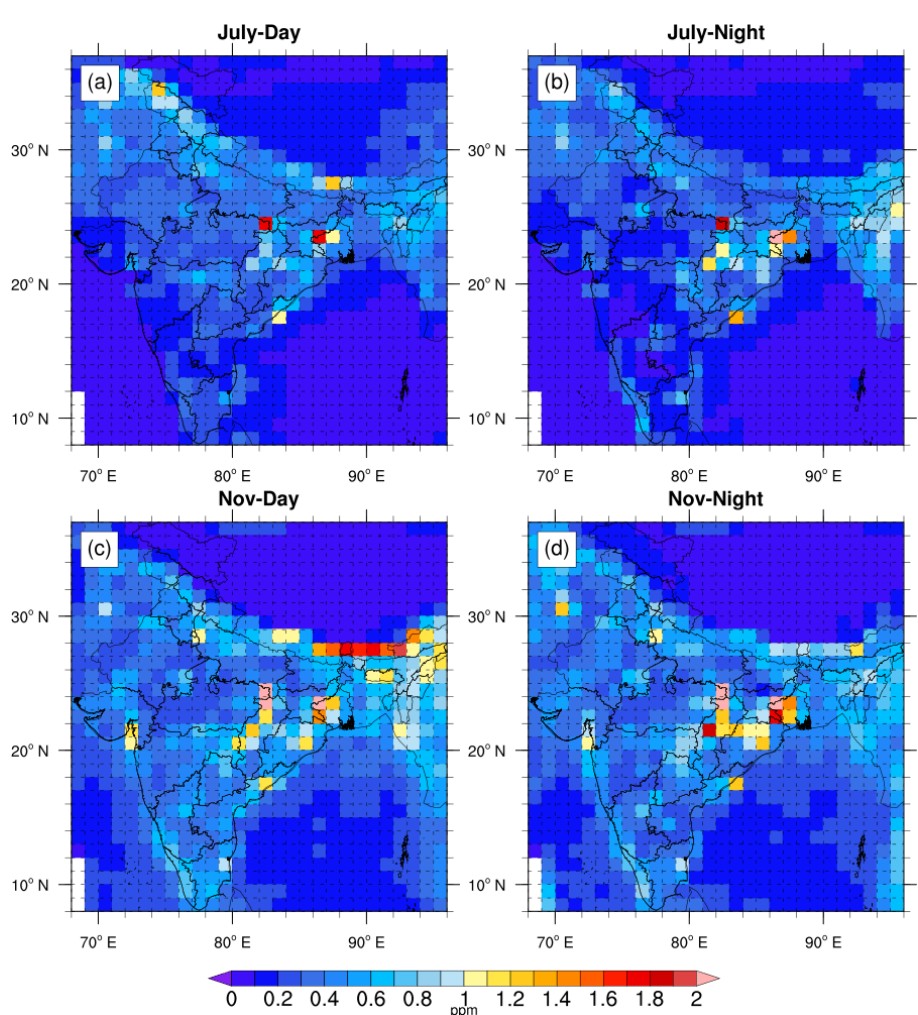

**Figure 5: Monthly averaged values of representation error estimated for column averaged CO$_2$ concentration over the region 8° N to 37° N and 68° E to 96° E during 2017. a) July daytime (11:30 to 16:30 local time) b) July nighttime (23:30 to 4:30 local time). c) November daytime. d) November nighttime.**







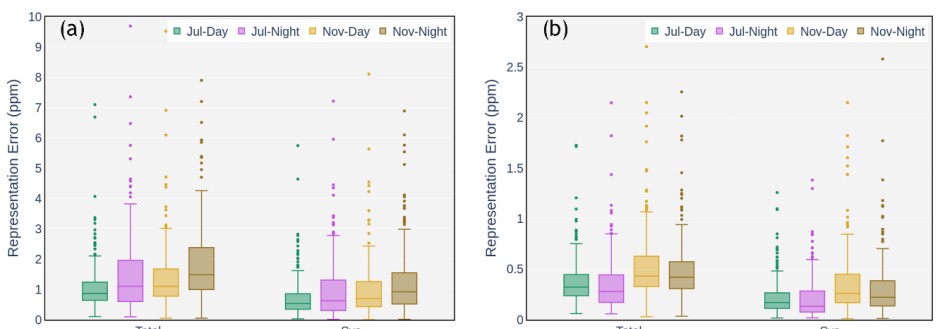

**Figure 6: Variability of derived representation error over India in July and November 2017 (both during daytime and nighttime). Boxes indicate the central 50%, the bar across the box is the median value, and the whiskers indicate the values between 5 and 95 percentiles. Individual data points shown are the outliers. a) Representation error estimated for the surface CO₂. b) Representation error estimated for the column averaged CO₂.**


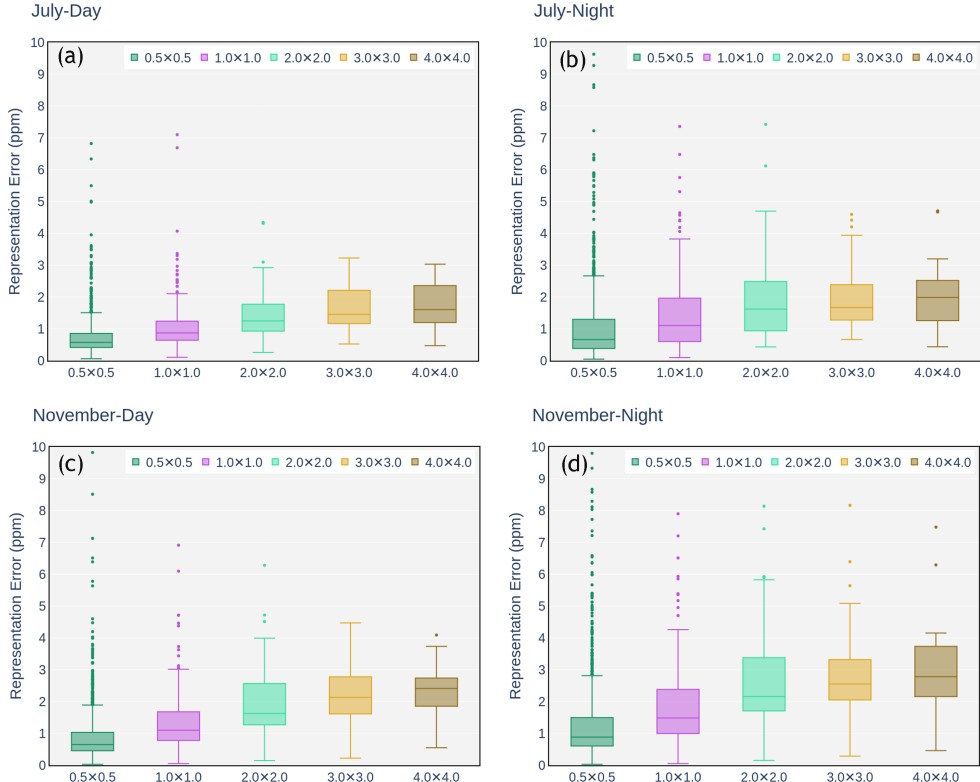

**Figure 7: Variability of derived surface representation error over India for different horizontal resolutions. Boxes indicate the central 50%, the bar across the box is median value, and the whiskers indicate the value between 5 and 95 percentiles. Individual data points shown are the outliers. a) Representation error estimated for July daytime. b) July nighttime. c) November daytime. d) November nighttime.**





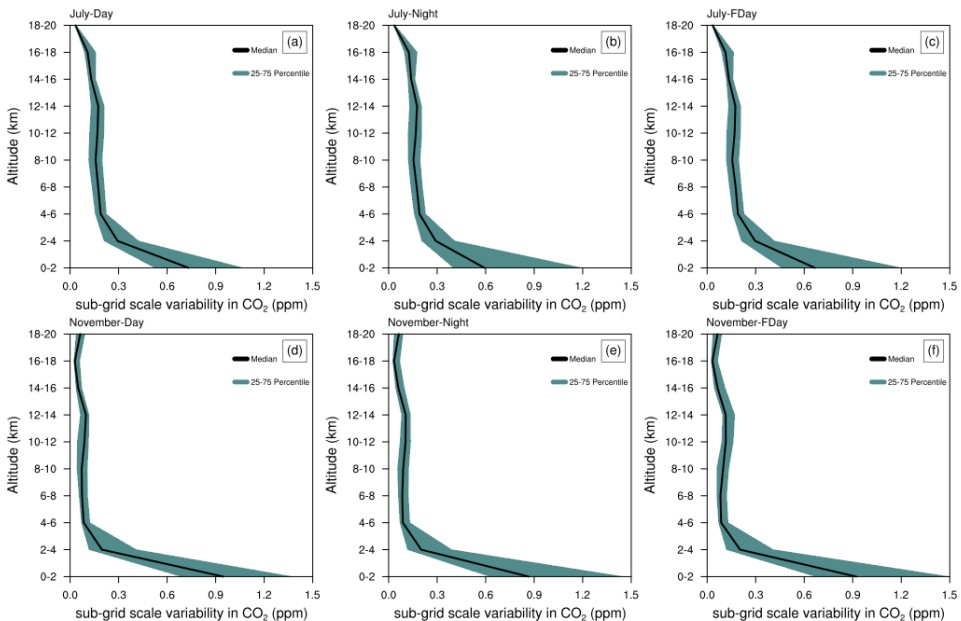


**Figure 8: Variability of representation error over India with altitude for July and November 2017. a) July daytime, b) July nighttime, c) July full time, d) November daytime, e) November nighttime, and f) November full time. Median values are plotted with black curves and the shaded region indicates 25 to 75 percentiles of data.**


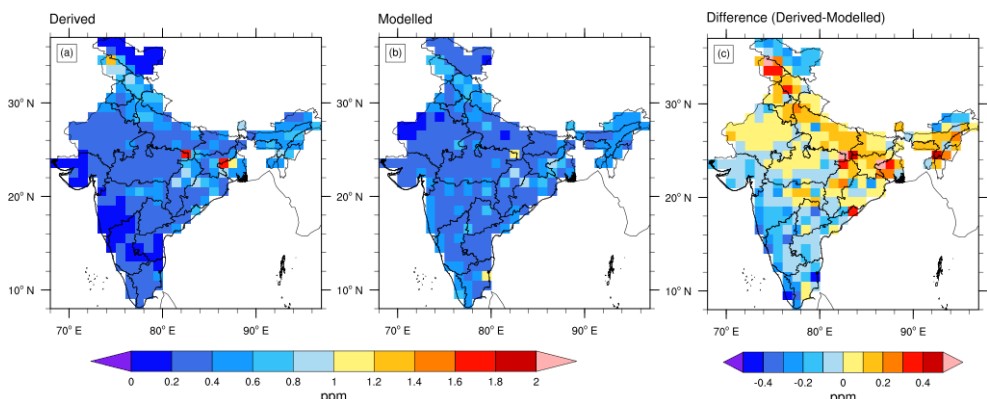

**Figure 9: Monthly averaged values of representation error estimated for column averaged $CO_2$ concentration during July daytime (11:30 to 16:30 local time) in 2017. a) Representation error derived from WRF-GHG simulations as explained in Sect. 2.3. b) Representation error calculated from the multivariate linear model as described in Sect. 3.5. c) Difference between (a) and (b).**




**Table 1: WRF-GHG Model setup**

| Domain | |
|---|---|
| Configuration | Single domain with horizontal resolution of 10 km; 39 vertical levels; 307 × 407 grid points |
| Vertical coordinates | Terrain-following hydrostatic pressure vertical coordinates |
| Basic equations | non-hydrostatic; compressible |
| Grid type | Arakawa-C grid |
| Time integration | 3rd order Runge-Kutta split-explicit |
| Spatial integration | 3rd and 5th order differencing for vertical and horizontal advection respectively; both for momentum and scalars |
| Timestep | 60 s |

| Physics schemes | |
|---|---|
| Radiation | Rapid Radiative Transfer Model (RRTM) for Longwave & Dudhia for shortwave |
| Microphysics | WSM 3-classic simple ice scheme |
| PBL | YSU |
| Surface layer | Monin-Obukhov |
| Land-surface | NOAH LSM |
| Cumulus | Grell-Freitas ensemble scheme |

| Emission fields | | | | | | |
|---|---|---|---|---|---|---|
| Flux type | Product | Version | Spatial resolution | Temporal resolution | Source/website | Reference |
| Anthropogenic Biomass burning Biospheric | EDGAR GFAS VPRM | v4.3 v1.2 | 10km 10km Adapted to model | Annual Daily Adapted to model | https://edgar.jrc.ec.europa.eu/ http://apps.ecmwf.int/datasets /data/cams-gfas/ | Crippa et al., (2018) Kaiser et al., (2012) Mahadevan et al., (2008) |

| Initial and Lateral Boundary conditions | | | | | | |
|---|---|---|---|---|---|---|
| Field | Product | Version | Spatial resolution | Temporal resolution | Source/website | Reference |
| Meteorology Tracer | ERA5 ECMWF /CAMS | n/a gqiq | 25km 50km | 1hour 6hour | https://cds.climate.copernicu s.eu/cdsapp#!/home http://atmosphere.copernicus .eu | Hersbach et al., (2020) Agustí-Panareda et al., (2019) |



**Table 2: Specifications of different global model products used in this study**

| Data availability | | | | | | |
|---|---|---|---|---|---|---|
| Product | Version | Spatial resolution | Vertical levels | Temporal resolution | Source/website | Reference |
| Carbon Tracker | CT2019B | 3 × 2 | 25 | 3 hours | http://carbontracker.noaa.gov | Jacobson et al., (2020) |
| CarboScope | s10oc_v2020 | 5 × 3.8 | 19 | 6 hours | http://www.bgc-jena.mpg.de/CarboScope/ | Rödenbeck et al., (2003) |
| LSCE | v18r3 | 3.7 × 1.8 | 39 | 3 hours | http://atmosphere.copernicus.eu | Chevallier et al., (2019) |
| LSCE | FT18r1 | 3.7 × 1.8 | 39 | 3 hours | http://atmosphere.copernicus.eu | Chevallier et al., (2019) |

| Data used in the inverse model simulations | | | | | | | | |
|---|---|---|---|---|---|---|---|---|
| Product | Version | Forward Model | Meteorology | Observation data | Anthropogenic emission fields | Biospheric emission | Fire emission | Oceanic emission |
| Carbon Tracker | CT2019B | TM5 | ECMWF | Ground based | Miller and ODIAC | CASA | GFED and GFED CMS | OIF and Takahashi et al., (2009) |
| CarboScope | s10oc_v2020 | TM3 | NCEP | Ground based | EDGAR | LPJ Biosphere Model | CDIAC | SOCAT |
| LSCE/ PyVar | v18r3 | LMDz6A | ECMWF | Ground based | EDGAR, CDIAC and GCP | ORCHIDEE 4.6.9.5 | GFED and GFAS | Denvil-Sommer et al., (2019) with updates described in Friedlingstein et al., (2019) |
| LSCE/ PyVar | FT18r1 | LMDz6A | ECMWF | Satellite (OCO-2 NASA) | EDGAR, CDIAC and GCP | ORCHIDEE 1.9.5.2 | GFED and GFAS | Landschutzer et al., (2018) |





**Table 3: Flux uncertainty over India calculated from the OSSE experiments using pseudo-observation network of surface observations. The time filter indicates the time of the data sampled for estimation of the scaling factors. Full day – 24 hours in each day; Daytime – 11:30 to 16:30 local time; Nighttime – 23:30 to 4:30 local time. * The fraction of uncertainty to the total NEE.**

| Month | Time filter | True flux, aggregated over India. $\sum_{k=1}^{K} \Phi_{true}$ (MtCO$_2$ per month) | Flux uncertainty $S_{rep}$ (MtCO$_2$ per month) In brackets: fraction of uncertainty* (%) |
|---|---|---|---|
| July | Daytime observations | | 38.59 (10.33) |
| July | Nighttime observations | -373.31 | 30.14 (8.07) |
| July | Full day observations | | 23.20 (6.21) |
| November | Daytime observations | | 18.42 (4.4) |
| November | Nighttime observations | -417.12 | 13.34 (3.1) |
| November | Full Day observations | | 13.48 (3.2) |

