# Peer review of "Towards monitoring CO2 source-sink distribution over India via inverse modelling: Quantifying the fine-scale spatiotemporal variability of atmospheric CO2 mole fraction"

_Atmospheric Chemistry and Physics, 2022_

## Author Response (AR1)

**Response to Reviewers' comments**

We are greatly thankful to Reviewers 1 and 2 for providing insightful comments on our manuscript. We have addressed all the comments, suggestions and concerns raised by the reviewers and incorporated associated modifications in the manuscript. Reviewers' comments (in black font) and Authors' responses (in blue regular font) are given below. Texts in the manuscript are given in *blue italic font*.

**R01:** Reviewer 1

**R02:** Reviewer 2

**AA:** All Authors
* * *
**R01**: The study by Thilakan et al. can be understood as a preparatory study for future inverse modelling of CO2 fluxes over India once an in-situ observation network with sufficient coverage is established. As a starting point, it investigates the spread in CO2 concentrations simulated by current state-of-the-art global CO2 data assimilation systems and then analyzes the impact of spatial representation errors in global models on the ability to inversely estimate CO2 fluxes over India. These representation errors are caused by the inability of coarse global models to resolve small-scale variations in CO2 due to variations in topography and biospheric and anthropogenic surface fluxes.

The present manuscript is much improved compared to an earlier version and is acceptable with minor revisions. In particular, the motivation of the work is much clearer now, although the individual elements still do not perfectly fit together. The analysis of differences in CO2 concentrations simulated by different global models is interesting (and in fact alarming), but this part is still only loosely connected to the core of the study, which is the analysis of representation errors. The conclusions section is much more compelling now than in previous versions.

Overall, the study provides a number of interesting analyses that will be valuable for future inversion studies over India and therefore deserves being published. These aspects include

- analysis of the factors contributing to small-scale CO2 variability
- quantification of corresponding representation errors in global models
- quantification of the impact of these representation errors on inverse estimation of CO2 fluxes
- presentation of an approach to reduce representation errors in global models, which accounts for the impact of sub-grid scale variability in orography and surface fluxes

I only have a number of minor comments that should be addressed before publication. I trust that other issues in terms of language and grammar will be corrected during the copy-editing phase.

**AA:** Thank you for appreciating our study. We are thankful for your careful review in helping us to improve the manuscript. We have addressed all your comments/suggestions and revised the manuscript accordingly. Please see the responses to the comments below.

**R01**:

- Line 57: It is true that the predictability of CO2 concentrations is better for windy situations, but in these situations CO2 will be dominated by the large background while the sensitivity to regional CO2 fluxes will be small

**AA:** We have modified the manuscript as follows:

L57-59: "*Strong wind normalizes other small-scale variations in observed concentration due to mixing. In such cases, the $CO_2$ variability is expected to be dominated by the variations in background concentration, hence the predictability can be higher during these conditions (Sarrat et al., 2007).*"

**R01**:

- L67: Replace "Further" by "Furthermore"
- L73: Change "over the regions" to "over other regions"
- L94: Replace "availability of radiation" simply by "radiation"
- L126-127: The sentence could be simplified to "The month of July represents a monsoon period when both biospheric and convective activity are strong".
- L129: "In contrast" would probably fit better than "On the other hand" Replace "the ways" by "possible ways"

**AA:** Done

**R01**:

- L206: I was unable to find information on this CAMS product (version 2.2.4). Is this a reanalysis product (like EGG4)? What is the original resolution of this product?
- L207-208: I doubt that the product has a 6-hour temporal and 0.5° spatial resolution. This is probably only the resolution at which the data was obtained, not the resolution of the original product.

**AA:** We used data taken from the greenhouse gas analysis experiment, which is one of the CAMS global products under active development for several years. More information is available at the CAMS website (see https://atmosphere.copernicus.eu/global-products and links therein). The analysis is conducted with the objective of providing realistic 3D fields of atmospheric GHG concentrations in dry air mole fractions for the global high-resolution forecast of GHGs, run at 9 km × 9 km resolution (Tco399L137). These products are in the developmental phase and not yet available to the general public (personal contact: Anna.Agusti-Panareda@ecmwf.int). While the original product is run at very high resolution, we have used the analysis 'suite' prepared for the forecasting runs ("gqiq") available on a horizontal grid equivalent to around 50km (but also with 137 vertical levels), and at 6 hourly intervals. We have modified the statement as follows:

L203-209: "*The initial and lateral boundary conditions of $CO_2$ tracers are obtained from the Copernicus Atmosphere Monitoring Service (CAMS) global greenhouse gas forecast products (currently in development, see Massart et al., 2016; Agusti-Panareda et al., 2019). Namely, we have used the dry air mole fractions of $CO_2$ from the CAMS greenhouse gas experiment analysis (gqiq), with a temporal resolution of 6-hour, horizontal resolution of*

*0.5° × 0.5° (original resolution 9km × 9km) and 137 vertical levels. Note that the CAMS product at 9 km × 9 km resolution is in the developmental phase and not yet available to the general public (personal contact: Anna.Agusti-Panareda@ecmwf.int).*"

**R01**:

- L221: Replace "during the year" by "for the year"
- L223: Delete "for the optimization" because it appears twice in the sentence.
- L238: Simply write "scales not captured" instead of "scales which could not be captured"
- L257-258: Change to "As space-borne instruments measure total columns rather than near-surface concentrations, we extend .."
- L261: Here and at several other places: It would be better to write "errors" rather than "error"
- L266: Change "difference in the sub-grid scale process" to "difference in sub-grid scale processes"
- L272-273: This could be written more elegantly as ".. which describes the systematic component of the representation error and provides important constraints for inversions .."
- L295: Replace "seasonal changes" by "seasonal differences" as you have only analyzed two different months but not the whole year.
- L298: Instead of "associated mesoscale activity." I suggest to write "associated convective activity that is only parameterized in global models".
- L372: Here and later in the same sentence: Replace "grids" by "grid cells"
- L382: Replace "using this approach can be inferred as the lower bound" by "using this approach may be considered as a lower bound"
- L389: Replace "is not sufficient to justify the model's performance" by "does not guarantee a good model performance"
- L404: Here and at a few other places "while" should be replaced by "when" (i.e. "when analyzing")

**AA:** Done.

**R01**:

- L436: The formulation "seasonally varying observations" makes no sense. An observation does not vary seasonally unless e.g. different instruments are used in different seasons.

**AA:** We have modified the statement as follows:

L440-441: *"quantify what would be typical representation errors associated with incorporating observations from different seasons into atmospheric models"*

**R01**:

- L441: Why should the satellite community gap-fill the observations? Standard satellite products do not provide gap-filled data.

**AA:** We do agree that standard satellite products do not provide gap-filled data. We attempt to indicate other applications that rely on modelled output when high precision observations are unavailable.

Revised as follows to make it clearer.

L444-447: "*Further, the seasonal spatial variability analysis of column averages can provide useful information to gap-fill the satellite-based products over India when large data gaps are present, which can be utilized for applications that do not demand high precision observations (e.g. Hammerling et al., 2012).*"

**R01**:

- L449: You should also cite Zellweger et al. (2016). https://amt.copernicus.org/articles/9/4737/2016/, which shows that the compatibility goal of WMO of 0.1 ppm is achievable with current instruments but is still quite challenging. You should also replace "less than 0.1 ppm" by "of the order of 0.1 ppm".

**AA:** Thank you. The statement is revised as follows:

- L453-455: "*In the case of high accuracy in situ measurements, the typical uncertainty for $CO_2$ measurements is of the order of 0.1 ppm (Andrews et al., 2014, Zellweger et al., 2016).*"

**R01**:

- L460: Replace "over the land and ocean boundary" by "at the boundary between land and ocean"

**AA:** Done.

**R01**:

- L486: The statement "The estimated column representation error is thus capable of causing significant biases in the satellite inferred CO2 fluxes over these regions" should be better backed up. How do these errors compare, for example, with the impact of XCO2 retrieval biases on regional CO2 fluxes discussed in Chevallier et al. (2007)? https://doi.org/10.1029/2006JD007375

**AA:** We have revised the sentence as follows:

L490-492: "*The estimated column representation errors over these regions are thus capable of causing significant biases in the satellite inferred $CO_2$ fluxes as regional biases of a few tenths of parts per million in column-averaged $CO_2$ can create a bias of a few tenths of a gigaton of carbon fluxes (Chevallier et al., 2007).*"

**R01**:

- L570: It should be "the dependence of the representation error on sigma_bio" rather than the other way round.

**AA:** Done

**R01**:

- L568: It is not quite clear whether the numbers presented here are based on an analysis of the variance or only of the standard deviations. This is important because only variances (and covariances) explained by different factors can be added up to explain the total variance. Standard deviations cannot be added up in this way.

**AA:** The numbers described here are based on squared correlation coefficients ($R^2$) between explanatory variable (e.g. $\sigma_{topo}$) and representation error. We have revised the manuscript as follows to enhance the clarity:

L558-559: "*We have estimated the statistical dependence ($R^2$) of representation error on topographic variability within the corresponding global climate models' grids to understand the relation between them*"

**R01**:

- L588: It is only the resolution of an atmospheric transport model that is a limitation, not the accuracy of the digital elevation model used to generate the model orography. Differences between different DEMs are typically very small.

**AA:** Revised as follows:

L595-598: "*In general, the above analysis underlines the need for using Digital Elevation Models (DEMs) at high resolution to take into account the terrain-induced mesoscale atmospheric flows adequately in atmospheric transport models. Further, the results indicate the importance of utilising high-resolution surface fluxes in atmospheric $CO_2$ simulations.*"

**R01**:

- L597 and following: How does the fact that only 35% of the area of India is covered by the 9 stations affect these results? This should be discussed in this paragraph.

**AA:** Included a discussion as follows:

L608-612: "*The spatial representativeness of measurement stations used in this study is assumed to cover only 35% of the country's total area (see Sect. 2.4). Consequently, the impact of representation error on flux uncertainty, as reported in this study, is an underestimation when measurements from more regionally representative sites or a dense observation network are utilized in inversions.*"

**R01**:

- L617: How good is the performance in terms of R-square? It would be good to add this information to be consistent with the statement on L613.

**AA:** Done. Revised as follows:

L627-630: "*Similarly, we have modelled the surface representation error using the linear model with these three explanatory variables and found that the proposed model could capture the derived surface representation error well ($R^2 = 0.89$) with a deviation less than 1 ppm in most of the regions (see Supplementary Fig. S7 and Supplementary Table S1 and S2).*"

**R01**:

- L624: Again, the accuracy of the DEM is not a limitation.

**AA:** Revised as follows:

L633-636: "*Applying this parameterization scheme to the specific problem requires a high-resolution map of the terrain and prior information on anthropogenic and biogenic fluxes as the uncertainties in the topography and surface fluxes can significantly impact flux estimation.*"

**R01**:

- L655: Comparing Figures 4 and 5 or looking at Figure 6, I don't see an order of magnitude difference in representation errors between surface and column CO2. The representation errors in column CO2 are in fact surprisingly large.

**AA:** Thank you. We made the correction as follows:

L667-669: "*Although the magnitude of the sub-grid variability of the total column is significantly smaller than the variability at the surface, the spatial pattern remains similar for both, owing to the dominance of surface heterogeneity in topography and fluxes.*"

**Response to Reviewer 2 comments**

**R02:** Review of the paper "Towards monitoring CO2 source-sink distribution over India via inverse modelling: Quantifying the fine-scale spatiotemporal variability of atmospheric CO2 mole fraction" by Thilakan et al. The authors have tried to estimate representation errors (REs) for model resolution, using a 10x10 km simulation by WRF-VPRM simulations. They also briefly discuss CO2 concentrations from global models. I have found this version of the ms reads better than the previous version. The paper still lacks clear direction. Impacts of nversion results are less than expected at the end. At the minimum the paper require major revisions before consideraion for publication in Atmos. Chem. Phys.

**AA:** Thank you for your comments and suggestions. We attempted to address all your comments and concerns. Please see our responses below. The manuscript is revised accordingly.

**R02:** Specific comments:

Line 145ff : I feel you do not need this paragraph, the introduction have talked about the rationale well. Go straight to model description

**AA:** Agree. Revised as follows:

L149-152: "*In the following subsections, we describe our high-resolution modelling system (Sect. 2.1), existing optimized global CO$_2$ simulations used in the study (Sect. 2.2), quantification of representation error (Sect. 2.3), and the observation system simulation experiment (OSSE) designed to estimate the impact of the derived sub-grid scale variations on flux estimations over India via inverse optimization (Sect. 2.4).*"

**R02:**

Line 214-2015: this is very strange, why cannot you do continuous simulation? you are loosing some 20% of computing time and may be some inconsistency in simulation with repeated initialisation + plus may be some discontinuity in transport.

**AA**: We followed this simulation strategy for taking advantage of assimilated meteorological fields (uses observation to optimize its output). When we perform continuous simulations, the skill of the models tends to decrease as the simulation time progresses (e.g. Agustí-Panareda et al., 2019; Ahmadov et al., 2012; Pillai et al., 2011). While the WRF model community has developed mechanisms designed to force some meteorological parameters towards either direct observations (aka *observational nudging*) or higher-level models assimilating observational data (aka *grid-nudging*), they only affect a limited number of meteorological variables and their impact on quality of GHG simulations is not fully known. Thus, we have decided to follow the already established frameworks of known performance in terms of GHG simulations.

Indeed, the continuous reinitialization of the transport does create inconsistencies, but these were estimated to be minor in the past sensitivity experiments. In order to further limit them, we have chosen the reinitialization time to be close to solar midnight over the study domain (18:00 UTC), which a) doesn't influence the $CO_2$ fluxes simulated by VPRM, and b) is often characterized by lower wind speeds, so that any potential inconsistencies at the moment of reinitialization should have only a limited impact on our simulated $CO_2$ fields.

**R02:**

Line 284: India specific study: Kulchala et al., Spatio-temporal variability of XCO2 over Indian region inferred from Orbiting Carbon Observatory (OCO-2) satellite and Chemistry Transport Model, Atmospehric Research, 269, 106044, 2022.

**AA:** Done. We have added the additional citation.

L97-99: "*Several studies showed that the monsoon system substantially impacts vegetation growth, generating distinct spatio-temporal patterns of the biogenic fluxes (e.g., Gadgil, 2003; Valsala and Maksyutov, 2013, Ravi Kumar et al., 2016, Kunchala et al., 2022).*"

L284-285: "*Due to the paucity of adequate ground-level observations over India, satellite observations play an essential role in the estimation of $CO_2$ fluxes via inverse modelling (e.g. Philip et al., 2022).*"

**R02:**

Line 299: there is a paper discussing these statements here (Patra et al., ACP, 2011)

**AA:** Done the additional citation.

L301-302: "*The presence of enhanced biospheric activity during July can reduce the $CO_2$ concentration in the lower troposphere (e.g. Patra et al., 2011).*"

**R02:**

Line 387ff: I still don't know what this means or if it is needed here? If true, why are you doing this exercise anyways? What are the relevance to your WRF simulation analysis? Any comparison or cross checking? I am still not convinced what the authors are aiming at with

this analysis (Figs. 2 & 3). there are measurements from CONTRAIL if you want to get a true picture of the global model uncertainties using model-observation comparison please check Patra et al. (2011) for a methodology. i think the readers need a bit more clarification about this analysis

**AA:** We have conducted the comparison of global model products to report the typical inter-model mismatches even at a monthly or annual scale (not to quantify the model-observation error which is beyond the scope of this study). The inter-model mismatches indicate the limitation in our understanding on representing (in existing models) the mechanisms governing $CO_2$ distribution over India. A part of these mismatches can be arisen due to the inability of coarse resolution global models to simulate small-scale processes that can lead to representation errors, biasing inverse estimation of $CO_2$ fluxes, if unaccounted. In such cases, the usability of dense observations (as a future scenario) would be reduced due to modelling uncertainties.

Relevance to WRF simulation analysis in this study: We have generated high-resolution WRF simulations to quantify the unresolved sub-grid variability (termed as representation errors) in coarse models. Our analysis concludes that the representation errors in coarse models and their impact on inverse estimation of fluxes cannot be ignored. Based on WRF simulation analyses, we also demonstrate an approach to minimize representation errors in global models, which accounts for the impact of sub-grid scale variability in orography and surface $CO_2$ fluxes.

To make it clearer, we have added/revised statements as follows:

L133-137: "*We have also utilized optimized $CO_2$ products at global scales to provide a more comprehensive overview of the typical mismatch among the existing model simulations over the Indian subcontinent even at monthly and annual scales. A part of these mismatches can be arisen due to the inability of coarse resolution global models to simulate the sub-grid scale processes which can lead to representation errors; thus, uncertainty in inverse estimations.*"

L391-393: "*Note that a mere agreement among the coarse models does not guarantee a good model performance over the region due to their plausibly large model errors in common and interdependency in terms of data sources.*"

L431-435: "*The extent of this unresolved variability in existing global models is further explored in Sect. 3.2. The spatial distribution of $CO_2$ concentration shows structural differences among these models (see Supplementary Fig. S3), which indicates a substantial knowledge gap in models for representing atmospheric $CO_2$ variability over the Indian subcontinent. As a consequence, the country's carbon budget estimations inferred via inverse modelling can be unreliable.*"

L656-659: *"For instance, we find that the unresolved variations (representation error) of global models with a spatial resolution of 1° × 1° can be ~1.5 ppm on average for the surface $CO_2$ that is even larger than the currently reported differences between global models (~1 ppm)."*

**R02:**

Line 417: is this a novel finding of this study for the this study area ??

**AA:** Based on our biosphere model simulations, here we discuss the seasonal variability in terrestrial carbon fluxes (note that we discussed about the monthly variations in $CO_2$ concentration). We are interested in deducing role of seasonal changes on generating $CO_2$ variability over India.

Revised as follows:

L419-422: "*The seasonal variation of monthly averaged $CO_2$ seen over the Indian subcontinent is mostly dominated by terrestrial carbon fluxes i.e., net ecosystem exchange (NEE) as seen from the VPRM simulations (see Supplementary Fig. S2) and as e.g. in Tiwari et al., 2013.*"

**R02:**

Line 422: what is the basis of plausibility ?

**AA:** The monsoon period is characterized by the presence of strong wind and convection, which would result in strong mixing of the trace gases in the lower troposphere. Following the above reasoning, the vertical gradient simulated by the LSCE model in the surface layers for this region can be highly unlikely.

Revised as follows:

L425-427: "*The strong vertical gradient in the surface levels as simulated by the LSCE model during the monsoon period is little plausible given the strong vertical mixing expected due to the presence of strong wind and convection.*"

**R02:**

Line 440: Why gap filling is needed ? You cannot replace measurement by model, if you believe the satellites are doing something right!

**AA:** We see the lack of clarity in the present statement. It was not our intention to say that model simulations can replace measurements. Instead, we would like to indicate that in some applications we can still rely on modelled output when high precision observations are unavailable.

Revised as follows to make it clearer.

L444-447: "*Further, the seasonal spatial variability analysis of column averages can provide useful information to gap-fill the satellite-based products over India when large data gaps are present, which can be utilized for applications that do not demand high precision observations (e.g. Hammerling et al., 2012).*"

**R02:**

Line 475ff: When you say representation error in CO2, I assume fossil-fuel component is also included in the analysis. I wonder why the cities or power plants hotspots are not revealed as the areas of high representation error ?

**AA:** Point emission sources like cities and power plants show very high representation errors both in July and November. Please see the following sentence in the manuscript.

L473-475: *"We can also find individual cells with high representation errors associated with point emission sources such as cities, mining sites, and coal-fired power plants at different parts of the domain."*

**R02:**

Line 481 : I was expecting about an order of magnitude lower representation error for XCO2, compared to REs in CO2 at 200 m. Can you give an equation how you calculated XCO2 ? also could you please show the REs in CO2 in the supplement, say at 2 km and 5 km altitude ? That will give an idea to the readers how REs propagate upwards

**AA:** We have calculated the $XCO_2$ using following equation (Pillai et al., 2010).

$$XCO_2 = \frac{\sum_{l=1}^{n}(m_l \cdot CO_{2,l})}{\sum_{l=1}^{n} m_l}$$

where, $m_l$ is the dry grid cell air mass and $CO_{2,l}$ is the mixing ratio at model level $l$ and $n$ is the number of levels used. We have excluded the topmost model level from the $XCO_2$ calculation.

We have estimated the variability of representation error with respect to altitude for both July and November (Please see Fig. 8. in the manuscript). Both of these months show a sharp reduction in representation errors as altitude increases.

**R02:**

Line 510ff: I like fig. 7. I think this is the most novel result of this study. But i am curious if you can quantify the REs arising from flux smoothing or transport error ? In TransCom continuous experiment Patra et al. (GBC, 2008) ran two sets of experiment to make such separation (please ref to their Fig. 8 associated discussion). Of course their method of estimation of REs is different from yours, but nevertheless the flux vs transport REs is an important information to derive from such OSSEs

**AA**: Thank you. Yes, it is of greater interest to quantify the influence of prior flux and transport errors separately on inverse estimations of fluxes. Fig.8 from Patra et al. (GBC, 2008) discussed model-observation mismatches (specifically correlation between modeled and observed $CO_2$) in a set of simulation experiments. But, as the reviewer rightly pointed out, this requires additional model simulations, new experimental set-up, observational data and analysis, and can be worth designing as an independent study. Decoupling the flux and transport errors from derived REs is not within the scope of our present study. Rather we are interested in factors influencing the sub-grid $CO_2$ variability in coarse models and demonstrating an approach to reduce these REs in coarse models without demanding a high-resolution simulation of transport and high computational costs.

**R02:**

Line 534: Again I would like to draw your attention to one paper here by Chandra et al. (ACP, 2017). Think there is not too much scopes or need for speculations. The Indian monsoon domain is well studied for dynamic and now for chemical species

**AA:** Done with additional citation.

L540-542: "*This may be associated with the presence of strong circulations in the upper troposphere and lower stratosphere, such as subtropical westerly jets or Asian summer monsoon anticyclone (e.g. Chandra et al., 2017).*"

**R02:**

Line 559ff: Not sure how you get these numbers for RE fractions. I am a bit concerned about the low contrast in REs with latitude or longitude over India, there are forest ecosystems, agricultural land, semi-arid land and deserts. I cannot explain the distributions of REs I see in Fig. 4 or Fig. 5.

**AA:** The spatial variability of representation error is more determined by the terrain heterogeneity and flux variability. High representation errors are found over the point source regions such as cities and coal-based power plants. Mountains and coastal regions also exhibit higher representation errors. Regions with strong biospheric activity (e.g. Western Ghats and North Eastern regions) show high representation error as well (please see Sect. 3.2.1).

Please see:

L455-456: "*A remarkable feature is the presence of very high representation error over North-East and Western Ghats regions, where the biosphere activity is very prominent.*"

L457-460: "*Also, we can find high representation error along the foothills of the Himalayas. In addition to the complex terrain, the region over the Ganges basin is characterized by increased anthropogenic activity, which contributes to a larger representation error surrounding this region*"

L473-475: "*We can also find individual cells with high representation errors associated with point emission sources such as cities, mining sites, and coal-fired power plants at different parts of the domain.*"

**R02:**

How good is the VPRM ? Any testing has been done? At the very least can you compare your results with global model results for seasonal cycles, e.g., in Fig. 2b ??

**AA:** Validation of the VPRM simulations is limited due to the lack of available observations over this region. But as we mentioned in the manuscript a number of studies which used VPRM for other regions around the world shows good prediction skills (Ahmadov et al., 2009; Pillai et al., 2011; Liu et al., 2018; Park et al., 2018). At the moment we are not able to robustly compare the results for seasonal cycles over India, as our high-resolution WRF simulations are only performed for two months (July and November), but a separate manuscript is under preparation to explore prior flux variability over Indian region using VPRM simulations. Please also note that the analysis here reports the percentage of uncertainty in flux estimation due to the estimated unresolved sub-grid variations in coarse models. i.e., the OSSE considers VPRM-derived fluxes are true fluxes and the percentage of uncertainty is influenced by the representation errors.

**R02:**

Line 596 : how can you explain higher CO2 sink in Nov compared to July over India (Fig. S2b suggest otherwise). also other published literature suggests higher uptake in July than November (Patra et a., ACP, 2011)

**AA**: The values reported in the manuscript are the Net Ecosystem Exchange (NEE) from VPRM model. The total Gross Primary Production (GPP) over India is -702.39 $MtCO_2$ in July and -677.71 $MtCO_2$ in November. We found higher respiration fluxes in July (329.08 $MtCO_2$) compared to November (260.58 $MtCO_2$).

**References**

Hammerling, D. M., Michalak, A. M., and Kawa, S. R.: Mapping of $CO_2$ at high spatiotemporal resolution using satellite observations: Global distributions from OCO-2, Journal of Geophysical Research., 117, D06306, https://doi:10.1029/2011JD017015, 2012.

Philip, S., Johnson, M. S., Baker, D. F., Basu, S., Tiwari, Y. K., Indira, N. K., Ramonet, M. and Poulter, B.: OCO-2 satellite-imposed constraints on terrestrial biospheric $CO_2$ fluxes over South Asia. Journal of Geophysical Research: Atmospheres, 127, e2021JD035035. https://doi.org/10.1029/2021JD035035. 2022.

Tiwari, Y. K., Revadekar, J. V., and Kumar, K. R.: Variations in atmospheric Carbon Dioxide and its association with rainfall and vegetation over India. Atmospheric Environment, 68, 45–51. https://doi.org/10.1016/j.atmosenv.2012.11.040, 2013.

---

## Author Response (AR2)

Dear Editor,

Thank you for accepting our manuscript for final publication, and also for giving a chance to respond to Reviewer 2's comments. We have attempted to address the comments by the reviewer and incorporated associated modifications in the manuscript.

Reviewer comments (in black font) and Authors' responses (in blue regular font) are given below. Texts in the manuscript are given in *blue italic font*.

**Response to Reviewer 2 comments**
* * *
**Reviewer:** I thank the authors for revising the manuscript by taking in to account most of my comments and those from the other reviewer. The article has improved, but I am still not fully satisfied with the explanation that global model intercomparison should be here and attribution of the simulation differences basically to some sort of representation error, per se, as addressed in this study which arise from model grid resolution. Models differ due to prior fluxes (note that no inversion actually have used data over India), transport uncertainty and resolution representation error. You are still not able to compare your model simulations with the global model results. But I believe this paper could be published as it highlight one of persisting problem in atmospheric modelling studies.

**All Authors:** Thank you for reviewing our revised manuscript and appreciating the relevance of our study. We wish to keep the global model intercomparison results and discussion in the manuscript for the following reasons:

We agree that the mismatch among the existing global model simulations can be due to differences in prior fluxes, transport, model configuration, and resolution. The above reasoning sets our rationale to quantify the inter-model errors at a monthly or annual scale, indicating the current knowledge gap that impacts the inverse estimation of $CO_2$ fluxes over India. Since some of these inter-model mismatches can arise due to unaccounted representation errors, we further quantified and compared the mismatches with the possible representation error in global model simulations with $1° \times 1°$ resolution over the Indian domain. The comparison shows that the estimated representation errors are significant, which need to be addressed (please see: L657-660: "*For instance, we find that the unresolved ... models (~1 ppm).*"). We also acknowledge that a mere agreement among the coarse models does not mean good models' performance due to the model errors in common and interdependency in terms of data sources (Please see: L392-394: "*Note that a mere ... data sources.*").

We have modified the manuscript as follows for more clarity.

L135-138: "*These inter-model mismatches arise due to various reasons such as differences in input datasets (e.g. prior fluxes), transport and model configuration. A part of these mismatches can also arise due to the inability of coarse resolution global models to simulate the sub-grid scale processes that can lead to representation errors.*"